Earth System
Dynamics
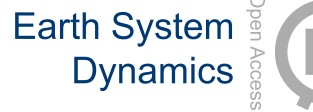

# Earth system modeling with endogenous and dynamic human societies: the copan:CORE open World–Earth modeling framework

**Jonathan F. Donges**[1,2,*], **Jobst Heitzig**[1,*], **Wolfram Barfuss**[1,3], **Marc Wiedermann**[1],
**Johannes A. Kassel**[1,4], **Tim Kittel**[3], **Jakob J. Kolb**[1,3], **Till Kolster**[1,3], **Finn Müller-Hansen**[1,5],
**Ilona M. Otto**[1], **Kilian B. Zimmerer**[1,6], and **Wolfgang Lucht**[1,7,8]

[1]Earth System Analysis and Complexity Science, Potsdam Institute for Climate Impact Research,
Member of the Leibniz Association, Telegrafenberg A31, 14473 Potsdam, Germany
[2]Stockholm Resilience Centre, Stockholm University, Kräftriket 2B, 114 19 Stockholm, Sweden
[3]Department of Physics, Humboldt University, Newtonstr. 15, 12489 Berlin, Germany
[4]Max Planck Institute for the Physics of Complex Systems, Nöthnitzer Straße 38, 01187 Dresden, Germany
[5]Mercator Research Institute on Global Commons and Climate Change (MCC),
EUREF Campus 19, Torgauer Straße 12–15, 10829 Berlin, Germany
[6]Department of Physics and Astronomy, University of Heidelberg,
Im Neuenheimer Feld 226, 69120 Heidelberg, Germany
[7]Department of Geography, Humboldt University, Unter den Linden 6, 10099 Berlin, Germany
[8]Integrative Research Institute on Transformations of Human-Environment Systems,
Humboldt University, Unter den Linden 6, 10099 Berlin, Germany
[*]The first two authors share the lead authorship.

**Correspondence:** Jonathan F. Donges (donges@pik-potsdam.de) and Jobst Heitzig (heitzig@pik-potsdam.de)

**Abstract.** Analysis of Earth system dynamics in the Anthropocene requires explicitly taking into account the increasing magnitude of processes operating in human societies, their cultures, economies and technosphere and their growing feedback entanglement with those in the physical, chemical and biological systems of the planet. However, current state-of-the-art Earth system models do not represent dynamic human societies and their feedback interactions with the biogeophysical Earth system and macroeconomic integrated assessment models typically do so only with limited scope. This paper (i) proposes design principles for constructing world–Earth models (WEMs) for Earth system analysis of the Anthropocene, i.e., models of social (world)–ecological (Earth) coevolution on up to planetary scales, and (ii) presents the copan:CORE open simulation modeling framework for developing, composing and analyzing such WEMs based on the proposed principles. The framework provides a modular structure to flexibly construct and study WEMs. These can contain biophysical (e.g., carbon cycle dynamics), socio-metabolic or economic (e.g., economic growth or energy system changes), and sociocultural processes (e.g., voting on climate policies or changing social norms) and their feedback interactions, and they are based on elementary entity types, e.g., grid cells and social systems. Thereby, copan:CORE enables the epistemic flexibility needed for contributions towards Earth system analysis of the Anthropocene given the large diversity of competing theories and methodologies used for describing socio-metabolic or economic and sociocultural processes in the Earth system by various fields and schools of thought. To illustrate the capabilities of the framework, we present an exemplary and highly stylized WEM implemented in copan:CORE that illustrates how endogenizing sociocultural processes and feedbacks such as voting on climate policies based on socially learned environmental awareness could fundamentally change macroscopic model outcomes.

Published by Copernicus Publications on behalf of the European Geosciences Union.

## 1 Introduction

In the Anthropocene, Earth system dynamics are equally governed by two kinds of internal processes: those operating in the physical, chemical and biological systems of the planet and those occurring in its human societies, their cultures and economies (Schellnhuber, 1998, 1999; Crutzen, 2002; Lucht and Pachauri, 2004; Steffen et al., 2018). The history of global change is the history of the increasing planetary-scale entanglement and strengthening of feedbacks between these two domains (Lenton and Watson, 2011). Therefore, Earth system analysis of the Anthropocene requires closing the loop by integrating the dynamics of complex human societies into integrated *whole* Earth system models (Verburg et al., 2016; Donges et al., 2017a, b; Calvin and Bond-Lamberty, 2018). Such models need to capture the coevolving dynamics of the social (the world of human societies) and natural (the biogeophysical Earth) spheres of the Earth system on up to global scales and are referred to as world–Earth models (WEMs) in this article. In pursuing this interdisciplinary integration effort, world–Earth modeling can benefit from and build upon the work done in fields such as social–ecological systems (Berkes et al., 2000; Folke, 2006) and coupled human and natural systems (Liu et al., 2007) research or large-scale behavioral land-use (Arneth et al., 2014; Rounsevell et al., 2014) and socio-hydrological modeling (Di Baldassarre et al., 2017). However, it emphasizes more the study of planetary-scale interactions between human societies and parts of the Earth's climate system such as atmosphere, ocean and the biosphere, instead of more local and regional-scale interactions with natural resources that these fields have typically focused on in the past (Donges et al., 2018).

The contribution of this paper is twofold: first, following a more detailed motivation (Sect. 1.1), general theoretical considerations and design principles for a novel class of integrated WEMs are discussed (Sect. 1.2) and WEMs are elaborated in the context of existing global modeling approaches (Sect. 1.3). Second, after a short overview of the copan:CORE open World–Earth modeling framework (Sect. 2), an exemplary full-loop WEM is presented and studied (Sect. 3), showing the relevance of internalizing sociocultural processes. Finally, Sect. 4 concludes the paper.

### 1.1 State of the art and research gaps in Earth system analysis

Computer simulation models are pivotal tools for gaining scientific understanding and providing policy advice for addressing global change challenges such as anthropogenic climate change or rapid degradation of biosphere integrity and their interactions (Rockström et al., 2009; Steffen et al., 2015). At present, two large modeling enterprises considering the larger Earth system in the Anthropocene are ma-

ture (van Vuuren et al., 2016). (i) Biophysical Earth system models (ESMs) derived from and built around a core of atmosphere–ocean general circulation models that are evaluated using storyline-based socioeconomic scenarios to study anthropogenic climate change and its impacts on human societies (e.g., representative concentration pathways, RCPs) (Stocker et al., 2013). (ii) Socioeconomic integrated assessment models (IAMs) are operated using storyline-based socioeconomic baseline scenarios (e.g., shared socioeconomic pathways, SSPs; Edenhofer et al., 2014) and evaluate technology and policy options for mitigation and adaption leading to different emission pathways. There is a growing number of intersections, couplings and exchanges between the biophysical and socioeconomic components of these two model classes for increasing their consistency (van Vuuren et al., 2012; Foley et al., 2016; Dermody et al., 2018; Robinson et al., 2018; Calvin and Bond-Lamberty, 2018).

However, the existing scientific assessment models of global change only include dynamic representations of the sociocultural dimensions of human societies to a limited degree – if at all (Fig. 1), i.e., the diverse political and economic actors, the factors influencing their decisions and behavior, their interdependencies constituting social network structures and institutions (Verburg et al., 2016; Donges et al., 2017a, b), and the broader technosphere they created (Haff, 2012, 2014). In IAMs, these sociocultural dimensions are partly represented by different socioeconomic scenarios (e.g., SSPs), providing the basis for different emission pathways. These are in turn used in ESMs as external forcing, constraints and boundary conditions to the modeled Earth system dynamics. However, a dynamic representation would be needed to explore how changes in the global environment influence these sociocultural factors and vice versa.

There are large differences in beliefs, norms, economic interests and political ideologies of various social groups and their metabolic profiles, which are related to their access and use of energy and resources (Fischer-Kowalski, 1997; Otto et al., 2019; Lenton et al., 2016; Lenton and Latour, 2018). Historical examples show that these differences might lead to rapid social changes, revolutions and sometimes also devastating conflicts, wars and collapse (Betts, 2017; Cumming and Peterson, 2017). In other cases, the inability to establish effective social institutions controlling resource access might lead to unsustainable resource use and resource degradation (see the discussion around the tragedy of the commons, Ostrom, 1990; Jager et al., 2000; Janssen, 2002). Climate change is a paradigmatic example of a global commons that needs global institutional arrangements for the use of the atmosphere as a deposit for greenhouse gas emissions if substantial environmental and social damage is to be avoided in the future (Edenhofer et al., 2015; Schellnhuber et al., 2016b; Otto et al., 2017).

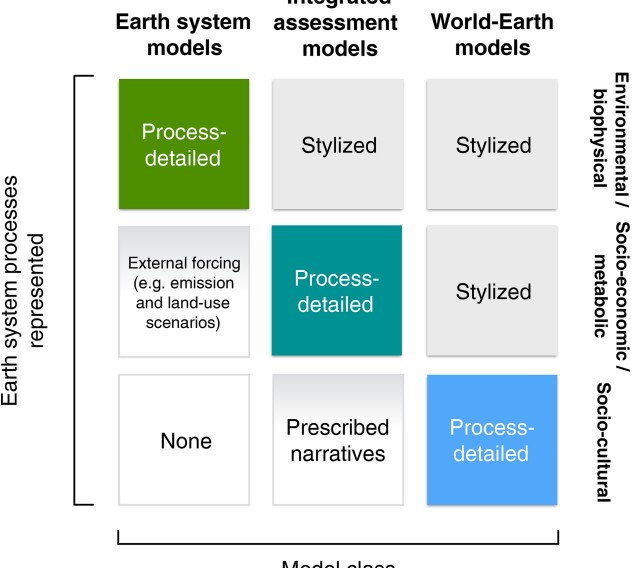

**Figure 1.** World–Earth models (WEMs) in the space of model classes used for scientific analysis of global change. It is shown to what degree current Earth system models, integrated assessment models and WEMs cover environmental or biophysical, socioeconomic or metabolic, and sociocultural processes. The term "process-detailed" indicates the types of Earth system processes that the different model classes typically focus on representing. However, also in these core areas the level of detail may range from very stylized to complex and highly structured.

In order to explore the risks, dangers and opportunities for sustainable development, it is important to understand how biophysical, socioeconomic and sociocultural processes influence each other (Donges et al., 2018), how institutional and other social processes function, and which tipping elements can emerge from the interrelations of the subsystems (Lenton et al., 2008; Kriegler et al., 2009; Cai et al., 2016; Kopp et al., 2016; Otto et al., 2020a). To address these questions, the interactions of social systems and the natural Earth system can be regarded as part of a planetary social–ecological system (SES) or world–Earth system, extending the notion of SES beyond its common usage to describe systems on local scales (Berkes et al., 2000; Folke, 2006). This dynamical systems perspective allows us to explore under which preconditions the maintenance of planetary boundaries (Rockström et al., 2009; Steffen et al., 2015), i.e., a Holocene-like state of the natural Earth system, can be reconciled with human development to produce an ethically defensible trajectory of the whole Earth system (i.e., sustainable development) (Raworth, 2012; Steffen et al., 2018).

## 1.2 World–Earth modeling: contributions towards Earth system analysis of the Anthropocene

To this end, the case has been made that substantial efforts are required to advance the development of integrated world–Earth system models for the study of the Anthropocene (Verburg et al., 2016; Donges et al., 2017a, b; Calvin and Bond-Lamberty, 2018). The need for developing such next-generation social–ecological models has been recognized in several subdisciplines of global change science dealing with socio-hydrology (Di Baldassarre et al., 2017; Keys and Wang-Erlandsson, 2018), land-use dynamics (Arneth et al., 2014; Robinson et al., 2018) and the globalized food–water–climate nexus (Dermody et al., 2018). While in recent years there has been some progress in developing stylized models that combine sociocultural with economic and natural dynamics (e.g., Janssen and De Vries, 1998; Kellie-Smith and Cox, 2011; Garrett, 2014; Motesharrei et al., 2014; Wiedermann et al., 2015; Heck et al., 2016; Barfuss et al., 2017; Nitzbon et al., 2017; Beckage et al., 2018), more advanced and process-detailed WEMs are not yet available for studying the deeper past and the longer-term Anthropocene future of this coupled system. The research program investigating the dynamics and resilience of the world–Earth system in the Anthropocene can benefit from recent advances in the theory and modeling of complex adaptive systems (Farmer et al., 2015; Verburg et al., 2016; Donges et al., 2017a, b; Calvin and Bond-Lamberty, 2018). When advancing beyond stylized modeling, a key challenge for world–Earth modeling is the need to take into account the agency of heterogeneous social actors and global-scale adaptive networks carrying and connecting social, economic and ecological processes that shape social–ecological coevolution (Otto et al., 2020b).

A number of new developments make it attractive to revisit the challenge of building such WEMs now. Due to the huge progress in computing, comprehensive Earth system modeling is advancing fast. And with the ubiquity of computers and digital communication for simulation and data acquisition in daily life (Otto et al., 2015), efforts to model complex social systems are increased and become more concrete. Recent advances, for example in the theory of complex (adaptive) systems, computational social sciences, social simulation and social–ecological system modeling (Farmer and Foley, 2009; Farmer et al., 2015; Helbing et al., 2012; Müller-Hansen et al., 2017; Schill et al., 2019) make it feasible to include some important macroscopic dynamics of human societies regarding, among others, the formation of institutions, values and preferences and various processes of decision-making in a model of the whole Earth system, i.e., the physical Earth including its socially organized and mentally reflexive humans. Furthermore, new methodological approaches are developing fast that allow representing crucial aspects of social systems, such as adaptive complex networks (Gross and Blasius, 2008; Snijders et al., 2010). Finally, initiatives such as Future Earth (Fu-

ture Earth, 2014), the Earth League (Rockström et al., 2014, https://www.the-earth-league.org/, last access: 1 April 2020) and the Open Modeling Foundation (Barton and The Open Modeling Foundation, 2019) provide a basis for inter- and transdisciplinary research that could support such an ambitious modeling program.

It is important to emphasize that despite these advances, integrated world–Earth modeling studies still face challenges particularly in the areas of selecting and managing the appropriate level of model complexity, mathematical representations of human behavior and social dynamics, costs of computation and model development, and data availability and consistency, as highlighted by a recent literature review (Calvin and Bond-Lamberty, 2018). While at least a subset of these challenges tends to apply to many other ambitious modeling projects in diverse fields, they have been used as a basis of criticism of past human-environment modeling exercises such as the classic WORLD3 model in the "Limits to growth" studies (Meadows et al., 1972). To address these challenges, as we detail in Sect. 2, world–Earth system modeling should be developed following a modular approach, allowing for the intercomparison of a diversity of modeling approaches and corresponding extensive robustness and uncertainty analyses (Verburg et al., 2016). Model types and complexity levels should be selected carefully depending on the research questions of interest (van Vuuren et al., 2016). Community development is needed to foster the necessary interdisciplinary collaboration and to develop common protocols and ontologies for data, model simulations and intercomparison projects (Otto et al., 2015; Verburg et al., 2016; Calvin and Bond-Lamberty, 2018; Barton and The Open Modeling Foundation, 2019).

### 1.2.1  Research questions for world–Earth modeling

We envision world–Earth modeling to be complementary to existing simulation approaches for the analysis of global change. WEMs are not needed where the focus is on the study of the biophysical and climatic implications of certain prescribed socioeconomic development pathways (e.g., in terms of emission and land-use scenarios), since this is the domain of Earth system models as used in the World Climate Research Programme's Coupled Model Intercomparison Project (CMIP) (Eyring et al., 2016) that provides input to the Intergovernmental Panel on Climate Change (IPCC) reports. Similarly, WEMs are not the tool of choice if the interest is in the normative macroeconomic projection of optimal socioeconomic development and policy pathways internalizing certain aspects of climate dynamics, e.g., the analysis of first- or second-best climate change mitigation pathways, since this is the domain of state-of-the-art integrated assessment models.

In turn, WEMs as envisioned by us here are needed when the research questions at hand require the explicit and internalized representation of sociocultural processes and their feedback interactions with biophysical and socioeconomic dynamics in the Earth system. In the following, we give examples of research questions of this type that could be studied with WEMs in the future, as they have been already elaborated in more detail by, e.g., Verburg et al. (2016), Donges et al. (2017a, b) and Beckage et al. (2018):

1. What are the relative strengths of feedback interactions between biophysical processes in the climate system and processes of decision-making and behavioral change in human societies (Calvin and Bond-Lamberty, 2018)? For example, what is their influence on the uncertainty of projected global warming under different emission and land-use scenarios (Beckage et al., 2018)?

2. What are the sociocultural, socioeconomic and environmental preconditions for sustainable development towards and within a "safe and just" operating space for humankind (Barfuss et al., 2018; O'Neill et al., 2018), i.e., for a trajectory of the Earth system that eventually neither violates precautionary planetary boundaries (Rockström et al., 2009; Steffen et al., 2015) nor acceptable social foundations (Raworth, 2012)?

3. A more specific example of the previous questions is: how can major socioeconomic transitions towards a decarbonized social metabolism, such as a transformation of the food and agricultural systems towards a sustainable, reduced-meat diet that is in line with recent recommendations by the EAT-Lancet Commission on healthy diets (Willett et al., 2019), be brought about in view of the strong sociocultural drivers of current food-related and agricultural practices and the reality of the political economy in major food-producing countries? And how would their progress be influenced by realized or anticipated tipping of climatic tipping elements like the Indian monsoon system `CE1` `TS1` (Wiedermann et al., 2019)?

4. Under which conditions can cascading interactions between climatic (e.g., continental ice sheets or major biomes such as the Amazon rain forest) and potential social tipping elements (e.g., in attitudes towards ongoing or anticipated climate change or eco-migration) be triggered and how can they be governed (Schellnhuber et al., 2016a; Steffen et al., 2018; Wiedermann et al., 2019)? What are implications for biophysical and social–ecological dimensions of Earth system resilience in the Anthropocene (Donges et al., 2017a)?

5. How do multilevel coalition formation processes (like the one modeled in Heitzig and Kornek (2018) assuming a static climate) interact with Earth system dynamics via changes in regional damage functions, mitigation costs, and realized or anticipated distributions of extreme events that drive changes in public opinions, which in turn influence the ratification of international

treaties and the implementation of domestic climate policies?

6. How do certain social innovations including technology, policies or behavioral practices diffuse in heterogeneous agent networks that could have global-scale impacts on planetary-boundary dimensions (e.g., Farmer et al., 2019; Tàbara et al., 2018; Otto et al., 2020a)? Which factors, such as network structure, information access as well as information feedback and update time, affect the innovation uptake? What are the impacts of a certain social innovation uptake on different agent groups (e.g., on agents with different economic, social or cultural endowment)? (Hewitt et al., 2019)

### 1.2.2 Design principles for world–Earth models

To address research questions of the kind suggested by the examples given above, we suggest that the development of WEMs of the type discussed in this paper could be guided by aiming for the following properties.

1. *Explicit representation of social dynamics*. Societal processes should be represented in an explicit, dynamic fashion in order to do justice to the dominant role of human societies in the Anthropocene. (In contrast, social processes occur typically non-dynamically in ESMs as fixed socioeconomic pathways and in IAMs as intertemporal optimization problems.)

   Social processes such as behavioral change as described by the theory of planned behavior (Beckage et al., 2018) or social learning (Donges et al., 2018) may be included in models via comparably simple equation-based descriptions. Yet more detailed WEMs should also allow for representations of the dynamics of the diverse agents and the complex social structure connecting them that constitute human societies, using the tools of agent-based and adaptive network modeling (Farmer and Foley, 2009; Farmer et al., 2015; Müller-Hansen et al., 2017; Lippe et al., 2019; Schill et al., 2019). The social sphere is networked on multiple layers and regarding multiple phenomena (knowledge, trade, institutions, preferences, etc.) and that increasing density of such interacting network structures is one of the defining characteristics of the Anthropocene (Steffen et al., 2007; Gaffney and Steffen, 2017). While there is a rich literature on modeling various aspects of sociocultural dynamics (e.g., Castellano et al., 2009; Snijders et al., 2010; Müller-Hansen et al., 2017; Schlüter et al., 2017), this work so far remains mostly disconnected from Earth system modeling (Calvin and Bond-Lamberty, 2018). Accordingly, more detailed WEMs should be able to describe decision processes of representative samples of individual humans, social groups or classes and collective agents such as firms, households or governments. This includes the representation

of diverse objectives, constraints and decision rules, differentiating, for example, by the agent's social class and function and taking the actual and perceived decision options of different agent types into account.

2. *Feedbacks and coevolutionary dynamics*. WEMs should incorporate as dynamic processes the feedbacks of collective social processes on biogeophysical Earth system components and vice versa. The rationale behind this principle is that the strengthening of such feedbacks is one of the key characteristics of the Anthropocene (Beckage et al., 2018; Calvin and Bond-Lamberty, 2018). For example, anthropogenic greenhouse gas emissions drive climate change, which acts back on human societies through increasingly frequent extreme events and may in turn change human behaviors relevant for these emissions. Moreover, the ability to simulate feedbacks is central to a social–ecological and complex adaptive systems approach to Earth system analysis. Capturing these feedbacks enables them to produce paths in coevolution space (Schellnhuber, 1998, 1999) through time-forward integration of all entities and networks allowing for deterministic and stochastic dynamics. Here, time-forward integration refers to simulation of changes in system state over time consecutively in discrete time steps, rather than solving equations that describe the whole time evolution at once as in intertemporal optimization.

3. *Nonlinearity and tipping dynamics*. WEMs should be able to capture the nonlinear dynamics that are a prerequisite for modeling climatic (Lenton et al., 2008; Schellnhuber et al., 2016a; Lenton et al., 2019) and social tipping dynamics (Kopp et al., 2016; Milkoreit et al., 2018; Otto et al., 2020a) and their interactions (Kriegler et al., 2009; Cai et al., 2016) that are not or only partially captured in ESMs and IAMs. This feature is important because the impacts of these critical dynamics are decisive for future trajectories of the Earth system in the Anthropocene, e.g., separating stabilized Earth states that allow for sustainable development from hothouse Earth states of self-amplifying global warming (Heitzig et al., 2016; Steffen et al., 2018).

4. *Cross-scale interactions*. Modeling approaches for investigating social–ecological or coupled human and natural system dynamics have already been developed. However, they usually focus on local or small-scale human–nature interactions (Schlüter et al., 2012). Therefore, such approaches need to be connected across scales and up to the planetary scale and incorporate insights from macro-level and global modeling exercises (Cash et al., 2006; Lippe et al., 2019; Ringsmuth et al., 2019).

5. *Systematic exploration of state and parameter spaces*. WEMs should allow for a comprehensive evaluation of

state and parameter spaces to explore the universe of accessible system trajectories and to enable rigorous analyses of uncertainties and model robustness. Hence, they emphasize neither storylines nor optimizations but focus on the exploration of the space of dynamic possibilities to gain systemic understanding. This principle allows for crucial Anthropocene Earth system dynamics to be investigated with state-of-the-art methods from complex systems theory, e.g., for measuring different aspects of the stability and resilience of whole Earth system states and trajectories (Menck et al., 2013; van Kan et al., 2016; Donges and Barfuss, 2017) and for understanding and quantifying planetary boundaries, safe operating spaces, and their manageability and reachability as emergent system properties across scales (Heitzig et al., 2016; Kittel et al., 2017; Anderies et al., 2019).

## 1.3 World–Earth models compared to existing modeling approaches of global change

It is instructive to compare WEMs more explicitly than above to the two dominant existing classes of global change models – Earth system models and integrated assessment models (van Vuuren et al., 2016) – in terms of the degree to which they represent biophysical, socio-metabolic or economic and sociocultural subsystems and processes in the world–Earth system (Fig. 1). Before discussing how model classes map to these process types, we describe the latter in more detail.

### 1.3.1 Basic process taxa in world–Earth models

Based on the companion article by Donges et al. (2018) that is also part of the special issue in *Earth System Dynamics* on "Social dynamics and planetary boundaries in Earth system modeling", we classify processes occurring in the world–Earth system as three major taxa that represent the natural and societal spheres of the Earth system as well as their overlap (Fig. 2). We give only a rough definition and abstain from defining a finer, hierarchical taxonomy, being aware that gaining consensus among different disciplines on such a taxonomy would be unlikely, and we thus leave the assignment of individual processes and attributes to a given taxon to the respective model component developers:

- *Environment (ENV; environmental, biophysical and natural processes).* The "environment" process taxon is meant to contain biophysical or "natural" processes from material subsystems of the Earth system that are not or only insignificantly shaped or designed by human societies (e.g., atmosphere–ocean diffusion, growth of unmanaged vegetation, and maybe the decay of former waste dumps).

- *Metabolism (MET; socio-metabolic and economic processes).* The "metabolism" process taxon is meant to contain socio-metabolic and economic processes from material subsystems that are designed or significantly shaped by human societies (e.g., harvesting, afforestation, greenhouse gas emissions, waste dumping, land-use change, infrastructure building). Social metabolism refers to the material flows in human societies and the way societies organize their exchanges of energy and materials with nature (Fischer-Kowalski, 1997; Martinez-Alier, 2009).

- *Culture (CUL; sociocultural processes).* The "culture" process taxon is meant to contain sociocultural processes from immaterial subsystems (e.g., opinion adoption, social learning, voting, policy making) that are described in models in a way abstracted from their material basis. Culture in its broadest definition refers to everything people do, think and possess as members of society (Bierstedt, 1963, p. 129). Sociocultural processes such as value and norm changes have been suggested to be key for understanding the deeper human dimensions of Earth system dynamics in the Anthropocene (Nyborg et al., 2016; Gerten et al., 2018)

### 1.3.2 Mapping model classes to Earth system processes

Earth system models focus on the process-detailed description of biogeophysical dynamics (e.g., atmosphere–ocean fluid dynamics or biogeochemistry), while socio-metabolic processes (e.g., economic growth, greenhouse gas emissions and land use) are incorporated via external forcing and sociocultural processes (e.g., public opinion formation, political and institutional dynamics) are only considered implicitly through different scenarios regarding the development of exogenous socio-metabolic drivers (Fig. 1). Integrated assessment models typically contain a more stylized description of biophysical dynamics, are process-detailed in the socio-metabolic or economic domains, and are driven by narratives such as the SSPs (O'Neill et al., 2017) in the sociocultural domain. In turn, WEMs could ultimately integrate all three domains with varying focus depending on the research questions of interest. The focus of current and near-future developments in world–Earth modeling would likely lie on the development of a detailed description of sociocultural processes because they are the ones where the least work has been done so far in formal Earth system modeling.

## 2 The copan:CORE open world–Earth modeling framework

Here we give a short overview of the world–Earth open modeling framework copan:CORE that was designed following the principles given above (Sect. 1.2) and is more formally described and justified in detail in the Supplement. It enables a flexible model design around standard components

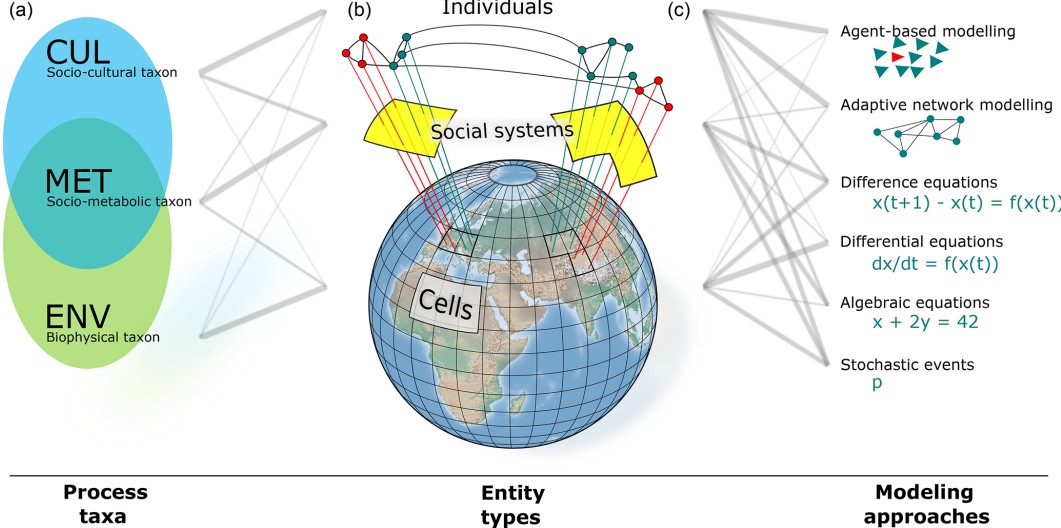

**Figure 2.** Overview of the copan:CORE open World–Earth modeling framework. The entities in copan:CORE models are classified by entity types (e.g., grid cell, social system, individual; see **b**). Each process belongs to either a certain entity type or a certain process taxon **(a)**. Processes are further distinguished by formal process types (see text for a list), which allow for various different modeling approaches **(c)**. Entity types, process taxa and process types can be freely combined with each other (gray lines). Thick gray lines indicate which combinations are most common. The copan:CORE framework allows us to consistently build world–Earth models across the spectrum from stylized and globally aggregated to more complex and highly resolved variants in terms of spatial and social structure. Hence, entity types, process taxa and types may or may not be present in specific models. For example, a stylized and globally aggregated model would describe the dynamics of the entity types "world" and "social system" and contain neither cells nor individual agents as entities.

and model setups that allows the investigation of a broad set of case studies and research questions using both simple and complex models. Its flexibility and role-based modularization support flexible scripting by end users, interoperability and dynamic coupling with existing models, and a collaborative and structured development in larger teams. copan:CORE is an open, code-based (rather than graphical) simulation modeling framework with a clear focus on Earth system models with endogenous human societies. In other words, it is a tool that provides a standard way of building and running simulation models without giving preference to any particular modeling approach or theory describing human behavior and decision-making and other aspects of social dynamics (Müller-Hansen et al., 2017; Schlüter et al., 2017). Different model components can implement different, sometimes disputed, assumptions about human behavior and social dynamics from theories developed within different fields or schools of thought. This allows for comparison studies in which one component is replaced by a different component modeling the same part of reality in a different way and exploring how the diverging assumptions influence the model outcomes.

All components can be developed and maintained by different model developers and can be flexibly composed into tailor-made models used for particular studies again by different researchers (Fig. 3). The framework facilitates the integration of different types of modeling approaches. It permits, for example, combining micro-economic models (e.g.,

of a labor market at the level of individuals) with systems of ordinary differential equations (modeling, for example, a carbon cycle). Similarly, systems of implicit and explicit equations (e.g., representing a multi-sector economy) can be combined with Markov jump processes (for example, representing natural hazards). It also provides coupling capabilities to preexisting biophysical Earth system and economic integrated assessment models and thus helps to benefit from the detailed process representations embedded in these models. Many of our design choices are based on experiences very similar to those reported in Robinson et al. (2018), in particular regarding the iterative process of scientific modeling and the need for open code, a common language for a broader community and a high level of consistency without losing flexibility. These features distinguish the copan:CORE modeling framework from existing modeling frameworks and platforms.

A model composed with copan:CORE describes a certain part of the world–Earth system as consisting of a potentially varying set of entities ("things that are", e.g., a spot on the Earth's surface, the European Union, yourself), which are involved in processes ("things that happen", e.g., vegetation growth, economic production, opinion formation) that affect entities' attributes ("how things are", e.g., the spot's harvestable biomass, the EU's gross product, your opinion on fossil fuels, the atmosphere–ocean diffusion coefficient) which represent the variables (including parameters) of a model. An attribute can have a simple or complex data type,

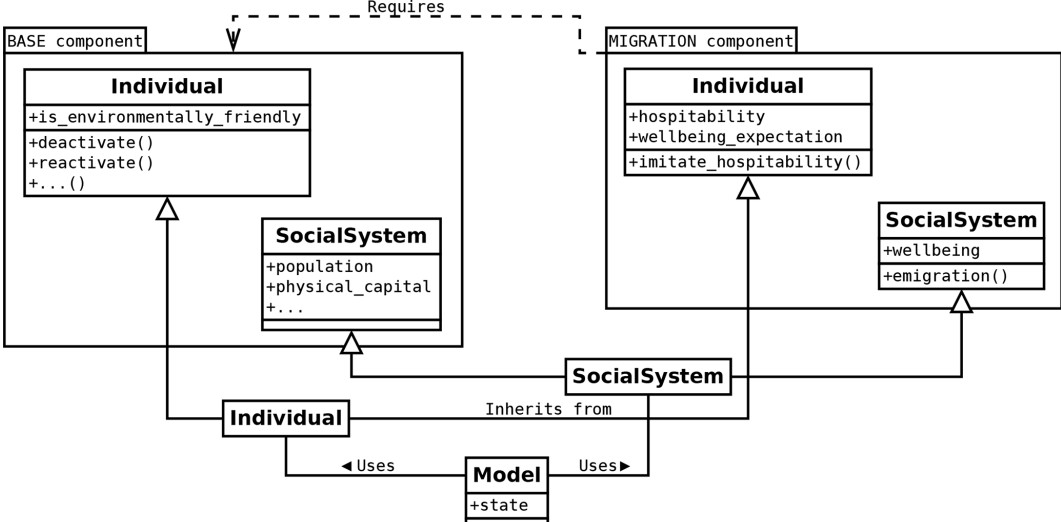

**Figure 3.** Model composition through multiple inheritance of attributes and processes by process taxa and entity types. This stylized class diagram shows how a model in copan:CORE can be composed from several model components (only two shown here: the mandatory component "base" and the fictitious component "migration") that contribute component-specific processes and attributes to the model's process taxa and entity types (only two shown here: "individual" and "SocialSystem"). To achieve this, the classes implementing these entity types on the model level are composed via multiple inheritance (solid arrows) from their component-level counterparts (so-called "mixin" classes).

e.g., representing a binary variable, a whole social network or, to facilitate interoperability and validation, a dimensional quantity with a proper physical unit.

Entities are classified by entity type (cell, social system, individual, etc.), processes by their formal process type (see below), and both are represented by objects in an object-oriented software design, currently using the Python programming language. Each process and each attribute belongs to an entity type or a process taxon (environmental, socio-metabolic, sociocultural). Currently, the following formal process types are supported, enabling typical modeling approaches:

- *ordinary differential equations* representing continuous time dynamics,

- *explicit* or *implicit algebraic equations* representing (quasi-)instantaneous reactions or equilibria,

- *steps* in discrete time representing processes aggregated at the level of some regular time interval or for coupling with external, time-step-based models or model components, and

- *events* happening at irregular or random time points, representing (e.g., agent-based and adaptive network components or externally generated extreme events).

Processes can be implemented either using an imperative programming style via class methods or using symbolic expressions representing mathematical formulae. co-

pan:CORE's modularization and role concept distinguish between

- *model components* developed by model component developers, implemented as sub-packages of the copan:CORE software package providing interface and implementation mixin classes for entity types and process taxa,

- *models* made from these by model composers, implemented by forming final entity types and process taxa from these mixin classes,

- *studies* by model end users in the form of scripts that import, initialize and run such a model,

- a *master data model* providing metadata for common variables to facilitate interoperability of model components and a common language for modelers, managed by a modeling board.

Entity types and their basic relations shipped with copan:CORE are the following:

- "world", representing the whole Earth (or some other planet).

- "cell", representing a regularly or irregularly shaped spatial region used for discretizing the spatial aspect of processes and attributes which are actually continuously distributed in space.

– "social system", such as a megacity, country or the EU. It can be interpreted as a human-designed and human-reproduced structure including the flows of energy, material, financial and other resources that are used to satisfy human needs and desires, influenced by the accessibility and use of technology and infrastructure (Fischer-Kowalski, 1997; Otto et al., 2020b), and may include social institutions such as informal systems of norms, values and beliefs and formally codified written laws and regulations, governance, and organizational structures (Williamson, 1998).

– "individual", representing a person, typically used in a network-theoretic, game-theoretic or agent-based component. In contrast to certain economic modeling approaches that use "representative" consumers, an individual in copan:CORE is not meant to represent a whole class of similar individuals (e.g., all the actual individuals of a certain profession) but just one specific individual. Still, the set of all individuals contained in a model will typically be interpreted as being a representative sample of all relevant real-world people. Each individual resides in a cell that belongs to a social system.

Figure 2 illustrates these concepts. Although there is no one-to-one correspondence between process taxa, entity types and modeling approaches, some combinations are expected to occur more often than others, as indicated by the thicker gray connections in Fig. 2. We expect environmental (ENV) processes to deal mostly with cells (for local processes such as terrestrial vegetation dynamics described with spatial resolution) and world(s) (for global processes described without spatial resolution, e.g., the greenhouse effect) and sometimes social systems (for mesoscopic processes described at the level of a social system's territory, e.g., the environmental diffusion and decomposition of industrial wastes). Socio-metabolic (MET) processes will primarily deal with social systems (e.g., for processes described at national or urban level), cells (for local socio-metabolic processes described with additional spatial resolution for easier coupling to natural processes) and world(s) (for global socio-metabolic processes such as international trade) and only rarely with individuals (e.g., for micro-economic model components such as consumption, investment or the job market). Sociocultural (CUL) processes will mostly deal with individuals (for "micro"-level descriptions) and social systems (for "macro"-level descriptions), and rarely world(s) (for international processes such as diplomacy or treaties). Other entity types such as firms, social groups or institutions can be added to the framework if needed.

## 3 Influence of social dynamics in a minimum-complexity world–Earth model implemented using copan:CORE

In this section, we present an illustrative example of a model realized with our framework. The example model was designed to showcase the concepts and capabilities of copan:CORE in a rather simple WEM, and its components were chosen so that all entity types and process taxa and most features of copan:CORE are covered. Although most model components are somewhat plausible versions of model components that can be found in the various literatures, the example model is intended to be a toy representation of the real world rather than one that could be used directly for studying concrete research questions. Likewise, although we show example trajectories that are based on parameters and initial conditions that roughly reproduce current values of real-world global aggregates in order to make the example as accessible as possible, the time evolutions shown may not be interpreted as any kind of meaningful quantitative prediction or projection.

In spite of this modest goal here, it will become obvious from the presented scenarios that including sociocultural dynamics such as migration, environmental awareness, social learning and policy making in more serious models of the global coevolution of human societies and the environment will likely make a considerable qualitative difference to their results and thus have significant policy implications.

The example model includes the following groups of processes: (1) a version of the simple carbon cycle used in Nitzbon et al. (2017) (based on Anderies et al., 2013) coarsely spatially resolved into four heterogeneous boxes; (2) a version of the simple economy used in Nitzbon et al. (2017) resolved into two world regions. The fossil and biomass energy sectors are complemented by a renewable energy sector with technological progress based on learning by doing (Nagy et al., 2013) and with human capital depreciation; and (3) domestic voting on subsidizing renewables and banning fossil fuels that is driven by individual environmental friendliness. The latter results from becoming aware of environmental problems by observing the local biomass density and diffuses through a social acquaintance network via a standard model of social learning (see, e.g., Holley and Liggett, 1975). These processes cover all possible process taxon interactions as shown in Table 1 and are distributed over six model components in the code as shown in Fig. 4.

We now describe the model components in detail. As many processes add terms to variables' time derivatives, we use the notation $\dot{X} + = Y$ to indicate this. The effective time evolution of $X$ is then determined by the sum of the individual processes given below.

**Table 1.** Possible classification of exemplary model processes by owning process taxon (row) and affected process taxon (column) (following the taxonomy developed in the companion paper Donges et al., 2018): environmental (ENV), social-metabolic (MET) and sociocultural (CUL).

| → | CUL | MET | ENV |
|---|---|---|---|
| CUL | social learning, voting | energy policy | environmental protection |
| MET | well-being | production, capital growth | extraction, harvest, emissions |
| ENV | well-being, awareness | resource availability | carbon cycle |

**Figure 4.** Components, entity types and processes of the example model. Each box represents a model component that contributes several processes (white bars) to different entity types and process taxa (differently hashed rectangles).

## 3.1 Entity types

The example model contains one world representing the planet, two social systems representing the Global North and South, four cells representing major climate zones: "boreal" and "temperate" belonging to the territory of the Global North and "subtropical" and "tropical" belonging to the Global South, and 100 representative individuals per cell, which form the nodes of a fixed acquaintance network.

## 3.2 Global carbon cycle

Our carbon cycle follows a simplified version of Anderies et al. (2013) presented in Nitzbon et al. (2017) with coarsely spatially resolved vegetation dynamics. On the world level, an immediate greenhouse effect translates the atmospheric carbon stock $A$ (initially 830 GtC) linearly into a mean surface air temperature $T = T_{\text{ref}} + a(A - A_{\text{ref}})$ (a process of type *explicit equation*) with a sensitivity parameter $a = 1.5\,\text{K}/1000\,\text{GtC}$ and reference values $T_{\text{ref}} = 287\,\text{K}$ and $A_{\text{ref}} = 589\,\text{GtC}$. There is ocean–atmosphere diffusion between $A$ and the upper-ocean carbon stock $M$ (initially 1065 GtC):

$$\dot{A} += d(M - mA), \quad \dot{M} += d(mA - M) \tag{1}$$

(processes of type "ODE"), with a diffusion rate $d = 0.016\,\text{yr}^{-1}$ and a solubility parameter $m = 1.5$. On the level of a cell $c$, $A$ and the cell's terrestrial carbon stock $L_c$ (initially 620 GtC for all four $c$) are changed by a respiration flow $\text{RF}_c$ and a photosynthesis flow $\text{PF}_c$:

$$\dot{A} += \text{RF}_c - \text{PF}_c, \quad \dot{L}_c += \text{PF}_c - \text{RF}_c. \tag{2}$$

The respiration rate depends linearly on temperature, which is expressed as a dependency on atmospheric carbon density $A/\Sigma$, where $\Sigma = 1.5 \times 10^8\,\text{km}^2$ is the total land surface area, so that

$$\text{RF}_c = (a_0 + a_A A/\Sigma)\,L_c, \tag{3}$$

with a basic rate $a_0 = 0.0298\,\text{yr}^{-1}$ and carbon sensitivity $a_A = 3200\,\text{km}^2\,\text{GtC}^{-1}\,\text{yr}^{-1}$. The photosynthesis rate also depends linearly on temperature (and hence on $A$) with an additional carbon fertilization factor growing concavely with $A/\Sigma$ and a space competition factor similar to a logistic equation, giving

$$\text{PF} = (l_0 + l_A A/\Sigma)\sqrt{A/\Sigma}\,(1 - L_c/k\Sigma_c)\,L_c, \tag{4}$$

with land area $\Sigma_c = \Sigma/4$, parameters $l_0 = 34\,\text{km}\,\text{GtC}^{-1/2}\,\text{yr}^{-1}$ and $l_A = 1.1 \times 10^6\,\text{km}^3\,\text{GtC}^{-3/2}\,\text{yr}^{-1}$,

and per-area terrestrial carbon capacity $k = 25 \times 10^3 \, \text{GtC}/1.5 \times 10^8 \, \text{km}^2$. Note that the linear temperature dependency and the missing water dependency, in particular, make this model rather stylized; see also Lade et al. (2018).

## 3.3 Economic production

As in Nitzbon et al. (2017), economic activity consists of producing a final good $Y$ from labor (assumed to be proportional to population $P$), physical capital $K$ (initially $K_{\text{North}} = 4 \times 10^{13}$, $K_{\text{South}} = 2 \times 10^{13}$, both given in units of USD), and energy input flow $E$. The latter is the sum of the outputs of three energy sectors, fossil energy flow $E_F$, biomass energy flow $E_B$, and (other) renewable energy flow $R$. The process is described by a nested Leontieff and Cobb–Douglas production function for $Y$ and Cobb–Douglas production functions for $E_F$, $E_B$ and $R$, all of them here on the level of a cell $c$:

$$Y_c = y_E \min\left(E_c, b_Y K_{Y,c}^{\kappa_Y} P_{Y,c}^{\pi_Y}\right), \quad E_c = E_{F,c} + E_{B,c} + R_c, \quad (5)$$

$$E_{F,c} = b_F K_{F,c}^{\kappa_F} P_{F,c}^{\pi_F} G_c^{\gamma}, \quad (6)$$

$$E_{B,c} = b_B K_{B,c}^{\kappa_B} P_{B,c}^{\pi_B} \left(L_c - L_c^p\right)^{\lambda}, \quad (7)$$

$$R_c = b_{R,c} K_{R,c}^{\kappa_R} P_{R,c}^{\pi_R} S_s^{\sigma}. \quad (8)$$

In this, $y_E = \text{USD} \, 147 \, \text{GJ}^{-1}$ is the energy efficiency, $G_c$ is the cell's fossil reserves (initially 0.4, 0.3, 0.2 and 0.1 $\times$ 1125 GtC in the boreal, temperate, subtropical and tropical cells), $L_c^p$ is the environmentally protected amount of terrestrial carbon (see below), $S_s$ gives the renewable energy production knowledge stock of the corresponding social system $s$ (initially $2 \times 10^{11}$ GJ), and $\kappa_\bullet = \pi_\bullet = \gamma = \lambda = \sigma = 2/5$ are elasticities leading to slightly increasing returns to scale. The productivity parameters $b_\bullet$ have units that depend on the elasticities and are chosen so that initial global energy flows roughly match the observed values: $b_F = 1.4 \times 10^9 \, \text{GJ}^5 \, \text{yr}^{-5} \, \text{Gt C}^{-2} \, \text{USD}^{-2}$ , $b_B = 6.8 \times 10^8 \, \text{GJ}^5 \, \text{yr}^{-5} \, \text{Gt C}^{-2} \, \text{USD}^{-2}$, and $b_{R,c} = 0.7, 0.9, 1.1$ and 1.3 times the mean value $b_R = 1.75 \times 10^{-11} \, \text{GJ}^3 \, \text{yr}^{-5} \, \text{USD}^{-2}$ in boreal, temperate, subtropical and tropical to reflect regional differences in solar insolation. As in Nitzbon et al. (2017), we assume $b_Y \gg b_B, b_F, b_R$ so that its actual value has no influence because then $K_{Y,c} \ll K_s$ and $P_{Y,c} \ll Y_s$. Furthermore, $K_{\bullet,c}$ and $P_{\bullet,c}$ are the shares of a social system $s$'s capital $K_s$ and labor $L_s$ that are endogenously allocated to the production processes in cell $c$ so that

$$K_s = \sum_{c \in s} \left(K_{Y,c} + K_{F,c} + K_{B,c} + K_{R,c}\right) \quad (9)$$

and similarly for its population $P_s$. The latter shares are determined on the social system level in a general equilibrium fashion by equating both wages (marginal productivity of labor) and rents (marginal productivity of capital) in all cells and sectors, assuming costless and immediate labor and capital mobility between all cells and sectors within each social system:

$$\partial y_E E_{F,c}/\partial P_{F,c} \equiv \partial y_E E_{B,c}/\partial P_{B,c} \equiv \partial y_E R_c/\partial P_{R,c} \equiv w_s \quad (10)$$

for all $c \in s$, and similarly for $K_{\bullet,c}$. The production functions and elasticities are chosen so that the corresponding equations can be solved analytically (see Nitzbon et al. (2017) for details), allowing us to first calculate a set of "effective sector or cell productivities" by a process of type *explicit equation* CE2 on the cell level, which are used to determine the labor and capital allocation weights $P_{\bullet,c}/P_s$ and $K_{\bullet,c}/K_s$, and then calculate output $Y_s$, carbon emissions, and all cells' fossil and biomass extraction flows in another process of type *explicit equation* on the social system level. Given the latter, a second process of type ODE on the social system level changes the stocks $A$, $G_c$ and $L_c$ for all cells accordingly.

## 3.4 Economic growth

Again as in Nitzbon et al. (2017), but here on the social system level, a fixed share $i$ (here 0.244) of economic production $Y_s$ is invested into physical capital $K_s$:

$$\dot{K}_s \mathrel{+}= i Y_s. \quad (11)$$

Capital also depreciates at a rate that depends linearly on surface air temperature to represent damage from climate change:

$$\dot{K}_s \mathrel{+}= -(k_0 + k_T (T - T_K)) K_s \quad (12)$$

with $k_0 = 0.1 \, \text{yr}^{-1}$, $k_T = 0.05 \, \text{yr}^{-1} \, \text{K}^{-1}$, and $T_K = 287$ K. In addition, renewable energy production knowledge $S_s$ grows proportional to its utilization via learning by doing:

$$\dot{S}_s \mathrel{+}= R_s. \quad (13)$$

Finally, we interpret $S_s$ as a form of human capital that also depreciates at a constant rate (due to forgetting or becoming useless because of changing technology, etc.):

$$\dot{S}_s \mathrel{+}= -\beta S_s, \quad (14)$$

with $\beta = 0.02 \, \text{yr}^{-1}$. Note that unlike in Nitzbon et al. (2017), we consider populations to be constant at $P_{\text{North}} = 1.5 \times 10^9$ and $P_{\text{South}} = 4.5 \times 10^9$ to avoid the complexities of a well-being-driven population dynamics component (which could, however, be implemented in the same way as in Nitzbon et al. (2017) on the social system level).

## 3.5 Environmental awareness

On the level of the culture process taxon, an "awareness updating" process of type "event" occurs at random time points with a constant rate (i.e., as a Poisson process, here with a rate of $4 \, \text{yr}^{-1}$), representing times at which many people become aware of the state of the environment, e.g., because of

notable environmental events. At each such a time point, each individual independently updates their environmental friendliness (a Boolean variable) with a certain probability. When individuals update, they switch from "false" to "true" with a probability $\psi^+$ depending on the terrestrial carbon density in their cell $c$, $\mathrm{TCD}_c = L_c/\Sigma_c$, given by

$$\psi^+ = \exp\left(-\mathrm{TCD}_c/\mathrm{TCD}^\perp\right), \tag{15}$$

and switches from true to false with a probability

$$\psi^- = 1 - \exp\left(-\mathrm{TCD}_c/\mathrm{TCD}^\top\right), \tag{16}$$

where $\mathrm{TCD}^\perp = 1 \times 10^{-5}$ and $\mathrm{TCD}^\top = 4 \times 10^{-5}$ are sensitivity parameters with $\mathrm{TCD}^\perp < \mathrm{TCD}^\top$ to generate hysteresis behavior. As a consequence, a fraction $L_c^{\mathrm{p}}$ of the terrestrial carbon $L_c$ is protected from harvesting for economic production. This fraction is proportional to the cell's social system's population share represented by those individuals which are environmentally friendly. The initial share of environmentally friendly individuals will be varied in the bifurcation analysis below.

## 3.6 Social learning

Similarly, on the culture level, "social learning" events occur at random time points with a constant rate (here $4\,\mathrm{yr}^{-1}$), representing times at which the state of the environment becomes a main topic in the public debate. At each such time point, each individual $i$ independently compares their environment with that of a randomly chosen acquaintance $j$ with a certain fixed probability (here $1/10$). $j$ then convinces $i$ to copy $j$'s environmental friendliness with a probability $\psi$ that depends via a sigmoidal function on the difference in logs between both home cells' terrestrial carbon densities:

$$\psi = 1/2 + \arctan\left(\pi\phi'\left(\log\mathrm{TCD}_j - \log\mathrm{TCD}_i - \log\rho'\right)\right)/\pi, \tag{17}$$

where $\phi' = 1$ and $\rho' = 1$ are slope and offset parameters. The underlying social network is a block model network in which each individual is on average linked to 10 randomly chosen others: 5 in the same cell, 3.5 in the other cell of the same social system and 1.5 in the other social system.

## 3.7 Voting on climate policy

Each (of the two) social systems performs general elections at regular time intervals (hence implemented as a process of type "step", here every 4 years) which may lead to the introduction or termination of climate policies. If at the time $t$ of the election, more than a certain threshold (here $1/2$) of the population is environmentally friendly, both a subsidy for renewables (here USD $50\,\mathrm{GJ}^{-1}$) is introduced and use of fossils is banned. This leads to a shift in the energy price equilibrium that determines the energy sector's allocation of labor

and capital, which then reads

marginal production cost of biomass energy

$=$ marginal production cost of renewable energy

$-$ renewable subsidy.

Conversely, if these policies are already in place but the environmentally friendly population share is below some other thresholds (here also $1/2$), these policies are terminated.

Note that we have chosen to model awareness formation and social learning in an agent-based fashion here mainly to illustrate that such an approach can easily be combined with other approaches in copan:CORE, not because we want to claim that an agent-based approach is the most suitable here. Indeed, one may well want to replace these two agent-based model components by equation-based versions which approximate their behavior in terms of macroscopic quantities (e.g., as in Wiedermann et al., 2015), and because of the modular design of copan:CORE, this can easily be done and the two model versions could be compared (nevertheless, this is beyond the scope of this paper).

## 3.8 Results

In order to show in particular what effect the inclusion of sociocultural processes into WEMs can have on their results, we compare two representative 100-year runs of the example model described above: one without the sociocultural processes environmental awareness, social learning, and voting (left panels of Fig. 5) and another with these processes included (right panels of Fig. 5). Both runs start in model year 0 from the same initial conditions and use the same parameters, which were chosen to roughly reflect real-world global aggregates of the year 2000 (see above). For the simulation without social processes (left panels of Fig. 5) both social systems ("Global North" as solid and "Global South" as dashed lines) initially rely on fossil energy in order to meet their energy needs, thus causing a rise in atmospheric and ocean carbon and a decline in fossil carbon stocks. Similarly both social systems initially rely heavily on energy from biomass, with the consequence of a reduction in terrestrial carbon. Due to the technology becoming competitive, the Global South changes its energy production to renewable energy comparatively early in the simulation, resulting in a fast fading out of biomass and fossils as an energy source. Due to its larger fossil reserves and lower solar insolation, the Global North takes 2 decades longer to make this switch. However, this delay in the Global North causes high atmospheric carbon, hence a high global mean temperature, which due to our oversimplified vegetation model makes the terrestrial carbon stock decline further even after biomass has been phased out as an energy source as well, recovering only much later (not shown). In both social systems, economic growth declines until the switch, then boosts and later declines again since neither population nor total factor productivity grow in

our model. Once the Global South switches to renewables, it hence overtakes the Global North, and this reversed inequality is then sustained as our model includes no trade, knowledge spillovers, migration or other direct interaction which would lead to economic convergence. Certainly, such results are not in themselves realistic (as this model does not intend to be) or transferable to real-world application. Future WEMs, therefore, should include such processes beyond pure economic ones in order to properly capture real-world–Earth dynamics; see the Supplement for some corresponding extensions of this model.

If social processes are considered, we obtain qualitatively similar but quantitatively different trajectories, e.g., in the right panels of Fig. 5, where we initially assume 40 % of all individuals are environmentally friendly. As before, both social systems initially rely on energy produced from fossils and biomass, but as biomass reduces terrestrial carbon density, environmental awareness makes some people environmentally friendly and this spreads via social learning. Once half of the population is environmentally friendly, the next elections in that social system bring a fossil ban and subsidies for renewables. This causes a slightly earlier switch to renewables than before, especially in the Global North (dashed lines in Fig. 5). This ultimately results in lower atmospheric and ocean carbon stocks, lower peak temperatures, less cumulative use of fossil fuels and a much faster recovery of terrestrial carbon.

copan:CORE further allows for a systematic investigation of the influence of individual parameters on the outcome of the simulation (e.g., along the lines of a bifurcation analysis). As an illustration of such an analysis we now vary the learning rate from $1/50\,\mathrm{yr}^{-1}$ (less than once in a generation) to $12\,\mathrm{yr}^{-1}$ (once every month) and compute the carbon stocks as well as the GDP per capita and the global mean temperature in model year 120 for an ensemble of 50 simulations per learning rate (Fig. 6) and the same initial conditions for all runs (we thus do not test for a possible multistability of the system).

For learning rates lower than $1\,\mathrm{yr}^{-1}$ (slow learning) the carbon stocks as well as the global mean temperature align well for the two simulation setups, i.e., the one with (scatter points) and without social processes (dashed lines). In contrast, for learning rates larger than $1\,\mathrm{yr}^{-1}$ (faster learning) the individuals become more capable of assessing the consequences of their behavior (in our case extensive biomass use) before the system has reached a state with low terrestrial and high atmospheric and ocean carbon stocks. As such, increasing the learning rate also causes an increase in the terrestrial carbon stock combined with a decrease in the atmospheric and ocean carbon stocks (in model year 120). This behavior is also reflected in the global mean temperature which decreases as the learning rate increases. Hence, with respect to the environment, social learning only has a positive effect if it happens at a sufficiently high rate (around once to more than once a year). It remains to note that learning rates have

in the past already been shown to have a profound impact on the state and dynamics of a coupled socio-ecological system, a feature that is recovered in our simple WEM as well (Wiedermann et al., 2015; Auer et al., 2015; Barfuss et al., 2017).

The metabolic variable GDP per capita interestingly already increases much earlier (i.e., for much lower learning rates than $1\,\mathrm{yr}^{-1}$) as compared to the changes in the environmental variables. This implies that for our specific WEM, social processes generally seem to foster the economy regardless of their actual rate. Furthermore we observe that the Global South shows an approximately 3 times higher GDP per capita than the Global North, which is caused by the earlier switch to renewable energies in that social system (see third row of Fig. 5). As already stated above, note again, that these results are not intended as a realistic projection of future trajectories of the Earth system, but are discussed here to showcase the capabilities of the copan:CORE framework.

Using the pycopancore reference implementation, running the above two simulations (Fig. 5) took 140 s (without sociocultural processes) and 290 s (including sociocultural processes) on an Intel Xeon E5-2690 CPU at 2.60 GHz. Since further performance improvements are desirable to support Monte Carlo simulations, we aim at a community-supported development of an alternative, more production-oriented implementation in the C++ language.

## 4 Conclusions

In this paper, we presented a simulation modeling framework that aims at facilitating the implementation and analysis of world–Earth (or planetary social–ecological) models. It follows a modular design such that various model components can be combined in a plug-and-play fashion to easily explore the influence of specific processes or the effect of competing theories of social dynamics from different schools of thought (Schlüter et al., 2017) on the coevolutionary trajectories of the system. The model components describe fine-grained yet meaningfully defined subsystems of the social and environmental domains of the world–Earth system and thus enable the combination and comparison of different modeling approaches from the natural and social sciences. In the modeling framework, different entities such as geographic cells, individual humans and social systems are represented and their attributes are shaped by environmental, socio-metabolic and sociocultural processes. The mathematical types of processes that can be implemented in the modeling framework range from ordinary differential and algebraic equations to deterministic and stochastic events. Due to its flexibility, the model framework can be used to analyze interactions at and between various scales – from local to regional and global.

The current version of the copan:CORE open modeling framework includes a number of tentative model components implementing, e.g., basic economic, climatic, biolog-

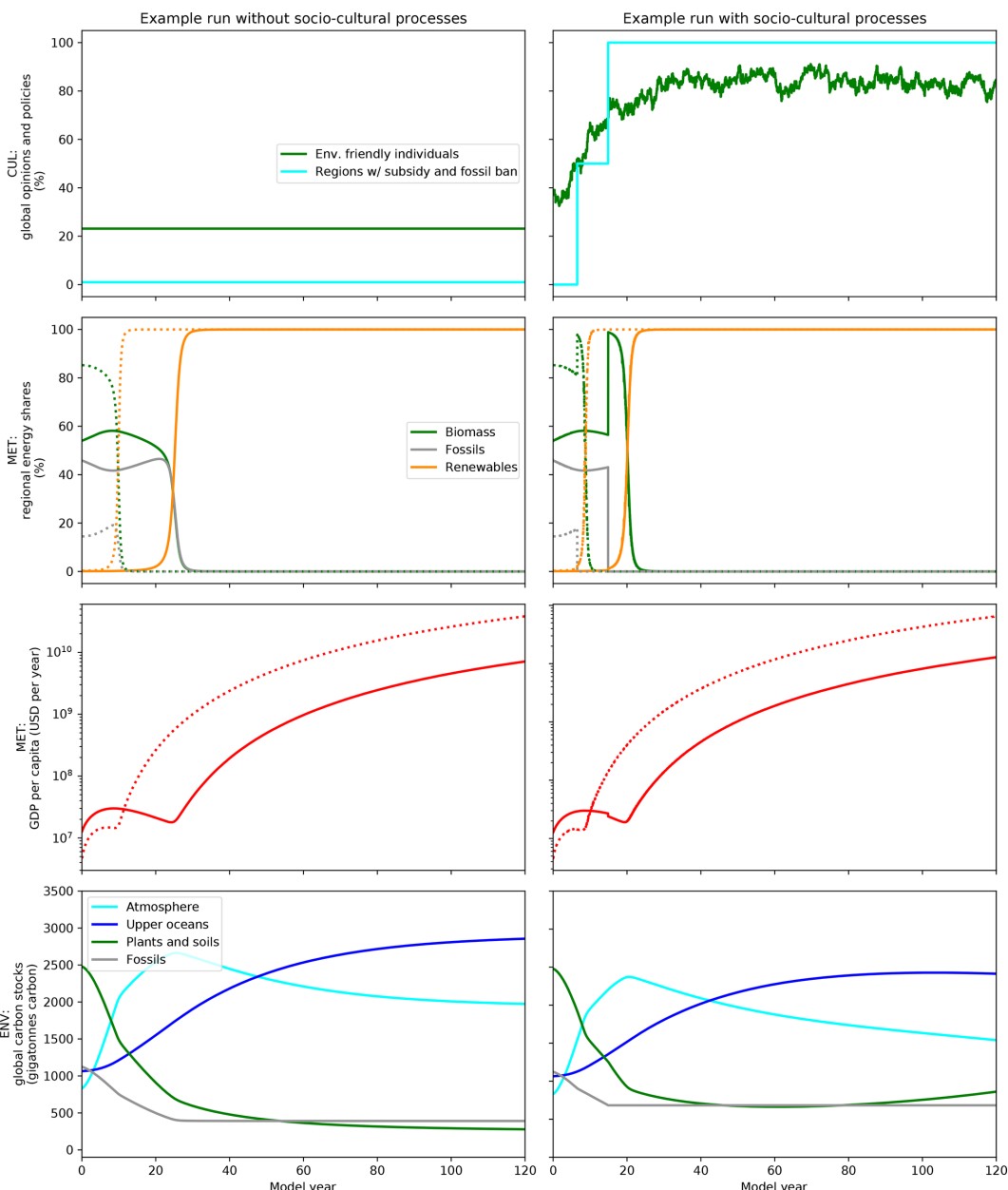

**Figure 5.** Two runs from a world–Earth model example: one without (left panels) and one with (right panels) the sociocultural processes of environmental awareness, social learning and voting included, showing different transient (and asymptotic) behavior. The top row shows variables related to the cultural process taxon, the second and third row those related to the metabolic process taxon and the bottom row those related to the environmental process taxon. Green, orange, cyan, blue and gray lines correspond to variables related to terrestrial carbon, renewables, atmospheric carbon, ocean carbon and fossils, respectively. In second and third row, dashed lines indicate variables associated with the "Global South", solid lines to the "Global North".

ical, demographic and social network dynamics. However, to use the modeling framework for rigorous scientific analyses, these components have to be refined, their details have to be spelled out and new components have to be developed that capture processes with crucial influence on world–Earth coevolutionary dynamics. For this purpose, various modeling approaches from the social sciences are available to be

applied to develop comprehensive representations of such socio-metabolic and sociocultural processes (Müller-Hansen et al., 2017; Schill et al., 2019, and references therein). For example, hierarchical adaptive network approaches could be used to model the development of social groups, institutions and organizations spanning local to global scales or the interaction of economic sectors via resource, energy and infor-

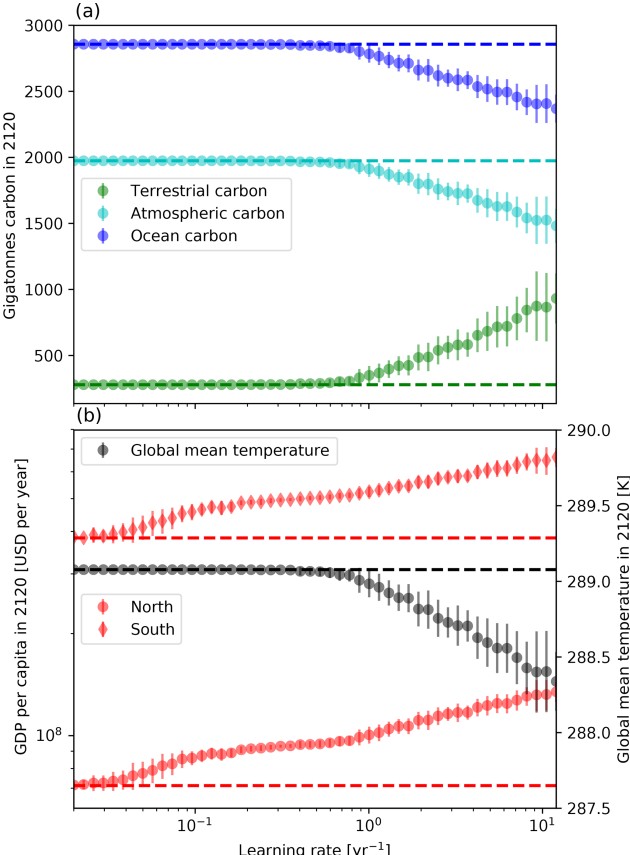

**Figure 6.** Dependency of some selected variables after 120 model years on the learning rate of environmental awareness. Scatter points denote (the average over 50) simulations with social processes, and error bars denote 1 standard deviation for each choice of learning rate. Dashed lines indicate the corresponding values for a simulation without social processes. Panel **(a)** shows the three environmental (non-fossil) carbon stocks; panel **(b)** shows the GDP per capita in the two social systems as well as the global mean temperature.

mation flows (Gross and Blasius, 2008; Donges et al., 2017a; Geier et al., 2019).

Making such an endeavor prosper requires the collection and synthesis of knowledge from various disciplines. The modular approach of the copan:CORE open modeling framework supports well-founded development of single model components, helps to integrate various processes and allows analyzing their interplay. To facilitate this, we envision an emergent community of modelers who contribute mature model components, composed models and variable definitions that add to a growing master component and model repository, and a master data model that are hosted within the open-source software repository (see below under "Code availability"), curated by a repository management board and cross-linked with platforms such as the CoMSES network (https://www.comses.net, last access: 1 April 2020). Complete models should also be contributed. This way, co-

pan:CORE could support the emergence of community standards for modeling coupled human–natural systems that have recently been demanded by many researchers (Barton and The Open Modeling Foundation, 2019). We therefore call upon the interdisciplinary social–ecological modeling community and beyond to participate in further model and application development to facilitate "whole" Earth system analysis of the Anthropocene.

**Code availability.** A Python 3.7.x implementation of the copan:CORE open World–Earth modeling framework, detailed documentation, a tutorial and the world–Earth model example are available at https://doi.org/10.5281/zenodo.3772751 (Heitzig et al., 2020).

**Supplement.** The supplement related to this article is available online at: https://doi.org/10.5194/esd-11-1-2020-supplement.

**Author contributions.** JFD and JH designed and coordinated the study. MW and JH performed model simulations and analyzed results. All other authors contributed to the writing of the paper and the discussion of results.

**Competing interests.** The authors declare that they have no conflict of interest.

**Special issue statement.** This article is part of the special issue "Social dynamics and planetary boundaries in Earth system modelling". It is not associated with a conference.

**Acknowledgements.** This work has been carried out within the framework of PIK's flagship project on Coevolutionary Pathways in the Earth system (copan, https://www.pik-potsdam.de/research/projects/activities/copan/copan-introduction, last access: 1 April 2020). We are grateful for financial support by the Stordalen Foundation via the Planetary Boundary Research Network (PB.net), the Earth League's EarthDoc program, the Leibniz Association (project DominoES), the European Research Council (ERC advanced grant project ERA), and the German Federal Ministry of Education and Research (BMBF, project CoNDyNet). We acknowledge additional support by the Heinrich Böll Foundation (WB), the Foundation of German Business (JJK), the Episcopal Scholarship Foundation Cusanuswerk (JK) and DFG/FAPESP (IRTG 1740/TRP 2015/50122-0, FMH). The European Regional Development Fund, BMBF and the Land Brandenburg supported this project by providing resources on the high-performance computer system at the Potsdam Institute for Climate Impact Research. We thank the participants of the three LOOPS workshops (https://www.pik-potsdam.de/research/projects/activities/copan/loops/loops-workshop-series, last access: 1 April 2020) in Kloster Chorin (2014), Southampton (2015) and Potsdam (2017) for discussions that provided highly valuable in-

sights for conceptualizing world–Earth modeling and the development of the copan:CORE open simulation modeling framework.

**Financial support.** The article processing charges for this open-access publication were covered by the Potsdam Institute for Climate Impact Research (PIK).

**Review statement.** This paper was edited by James Dyke and reviewed by Brian Dermody, Carsten Lemmen, and Axel Kleidon.

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

## Remarks from the language copy-editor

## Remarks from the typesetter