# Peer review of "Earth system modeling with endogenous and dynamic human societies: the copan:CORE open World-Earth modeling framework"

_Earth System Dynamics, 2017_

## Referee Comment (RC1) · B.J. Dermody (Referee) · 11 Apr 2018

*Review of* **Earth system modelling with complex dynamic human societies: thecopan:CORE World-Earth modeling framework by Donges and Heitzig et al. (2018)**

The authors present a modelling framework for a new generation of Earth System Models that they term World Earth Models (WEM). The paper presents their theoretical framework for capturing environmental, cultural and what they term, social metabolism processes in a linked model. They then provide details of the software package copan:CORE, which builds on their theoretical framework and is implemented in Python language.

It is this reviewers opinion a new generation of Earth System Models is urgently needed to capture complex dynamics between humans and the environment and this paper is an important first step in attempting to implement a modelling framework for a WEM. However, I would like to see more argumentation for the development of the theoretical framework they set out as well as clearer description of the model implementation, with consistency between the description of the theoretical framework and the model implementation framework. In addition, I suggest some structural changes to the paper.

**Paper Structure**

I think the paper could benefit from a slight restructuring for sections 1-3. The introduction introduces many terms without explanation or explanation comes in section 2 and 3. One important example of this is the term social-metabolism. This is later defined along with the other theoretical framings of the Earth System: environment and culture. You should introduce this framing earlier.

So, I would recommend starting with a shorter introduction with section 1.1 outlining the current state of modelling earth system processes, the shortcomings of these approaches and the motivation for a new framework. Then section 1.2 outline briefly, and in a language that users can follow (so if you introduce a new term such as social-metabolism, explain it), your theoretical framework, what problems it addresses and how it is implemented. Then in section 2 outline the theoretical framework in more detail. Crucial here is to motivate your reasoning behind the choices you make. This is not always clear in the discussion manuscript (Theoretical reasoning behind framework).

Then section 3 outline how the model is implemented. It should be very clear how the theoretical framework links with the implementation framework. This is not currently clear to me yet. Figure 2 is helpful, but I would like to see then how that relates to model framework structure: i.e. a figure like figure 4 but then capturing what is shown in figure 2. Importantly, if you keep more consistency between the theoretical framework structure and the implementation structure, then readers and users will be more easily able to follow what you have done.

**Theoretical reasoning behind framework**

Page 4 Line 10: The planetary boundaries concept has come in for some criticism lately (Montoya et al. 2018). A model framework such as this can potentially explain how planetary boundaries emerge through cross-scale human-environment interactions. It would be good to explain shortly how such a model framework could illuminate how we can understand how global planetary boundaries link across scales, keeping in mind the criticisms of the concept.

Page 4 Line 25 "environmental and societal processes should be described on similar levels of complexity" – sounds good but why? And what does that mean in reality? A tree and a person is equivalent? A country and an ecosystem equivalent? If so, what is the theoretical grounds for that?

Page 4 line 30-33, page 5 line 0-5 This seems all reasonable but why? And what is your grounding for these statements? In addition, there is a large body of work on applying agent-based models in the socio-cultural domain, which seems to have been ignored here. If you want to capture that, then you should demonstrate that you are aware of this literature and have considered it, including the many pitfalls of applying agent-based models to social systems. Also relates to the statement on Page 6, Line 13-15. There has been extensive work on formal modelling of socio-cultural processes. See Netlogo References for example: https://ccl.northwestern.edu/netlogo/references.shtml

Page 5 line 9-10 Outline why it is important to capture tipping points. This should also be covered in the intro when discussing shortcomings of existing models.

**Coupling or not?**

It is not clear to me whether the copan:CORE framework is designed to couple to other models such as LPJ-Guess and IAMs that you mention or if it is a standalone model with different modules or both? I.e. can external models can be modules within the copan:CORE framework? I would encourage you to outline this in more detail and with more prominence in the paper as a lot of the community are interested in a framework for coupling existing models that can incorporate the kind of dynamics you set out to include.

**Model description**

Generally, I find the model description too vague to know what can and can't be done with it. For instance, it is mentioned you can model resources flows and migration with it. How would this be implemented? Perhaps a few simple examples of specific model frameworks would help the reader understand what copan:CORE can and cant do. E.g. explain how would you use the framework to capture the relation between migration and drought or how tragedy of the commons scenarios emerges within a river catchment?

**Specific Comments**

Page 1 line 1 – 3.  This is quite a vague opening sentence. I would drop it. Start with: we introduce….(the abstract is already quite long)

Page 1 line 5: Not clear what is meant by user roles. Can you be more explicit, especially since this is the abstract.

Page 1 Line 14: I wouldn't include social metabolism in the abstract. Not a widely know term.

Page 3 line 30-35 Is this an agent-based model? From the abstract and introduction, I thought it was more than that. However, this concluding paragraph makes the reader think that you are going to introduce an ABM. If you view it as an ABM, fine. But then state that clearly in the abstract.

Page 5 line 8 what is time forward integration?

Page 7 Figure 2. While I like this figure, it could be clearer. It's not clear to me how each of the elements relate. Is each level of the network equivalent to Cul, Met and Env and are they then equivalent to the network on the right? It could be simpler to just show the central image (entity types) in one figure in the new intro section 1.2, for example. I like the way you show the different modelling approaches but it isn't clear with the network image how they relate or how cul,met, env relate.

Section 3.1 is very clear.

Page 10 line 4-5 delete "maybe changing numbers and"

Page 10 line 12 instead of following entity types, write entity types outlined in Section 3.2

Page 10 line 20: Give an example, such as countries to clarify

Page 11 line 4: human-designed, human-reproduced

Page 12 Line 6 – 30 Introduce this earlier in the manuscript. See my comments on structure

Page 16 Line 4: Examplary is not a word. It appears to be an obsolete form of exemplary which means "perfect".

A couple of important references "in prep". Try to find pre-existing publications to support arguments in addition to these where possible.

Page 22: Figure nested within references

**References**

Montoya, J.M., Donohue, I., Pimm, S.L., 2018. Planetary Boundaries for Biodiversity: Implausible Science, Pernicious Policies. Trends in Ecology & Evolution 33, 71–73. https://doi.org/10.1016/j.tree.2017.10.004

---

## Referee Comment (RC2) · C. Lemmen (Referee) · 17 May 2018

**1 General comments**

This manuscript by Donges and colleagues introduces the core technology and concept behind a new software tool called "copan", that should serve as "a framework for developing, composing and running World-Earth models". The authors motivate the development of such World Earth Models (WEM) that encompass dynamic descriptions of both the anthroposphere as well as the Earth System, they contrast WEM to integrated assessment and Earth System models, they describe the concepts of the

developed software package pycopancore and they show simple example applications of the software.

The contribution is within the scope of the Special Issue "Social dynamics and planetary boundaries in Earth system..." in Earth System Dynamics, although the preferred outlet for this kind of technical model description could also be Geoscientific Model Development. The novelty of the approach is the complexity of a World model combined with a stylized version of an Earth model; the innovation is in the open framework and theoretical embedding of the World Earth Model approach.

The paper is overall well written, but suffers from resilience theory and technical jargon, which should be reduced to address a wider readership. Figures are appropriate but they are of mixed graphical quality and accessibility and should be improved on. Tables are appropriate throughout; code examples examples are useful but in need of better quality. The supplementary material is well presented and useful.

The theory-laden motivation somewhat contrasts with the very technical model description. Reviewer one already remarked on the need for better embedding of these two major perspectives the manuscript assumes. I agree with that assessment, but for brevity I will concentrate in my detailed review below on other aspects of the manuscript. A major missing part is a description of how the presented copan:CORE framework fits into and operates with much of the existing coupling and model infrastructures in Earth and Social sciences; claims to interoperability, modularity and flexibility remain unsubstantiated.

I recommend that this paper is published after substantial revisions.

[Figure]

**2  Title, Abstract and related parts of Introduction**

**title**  There is an inconsistency in the spelling of "modelling" right in the title. Also, consider to spell out WEM as World Earth Model without hyphens; carefully consider lowercase/uppercase for "Model" in WEM. Nowhere in the paper the authors motivate the naming "copan:CORE"; please add a sentence on this naming and add to a table of abbreviations, if any of this is an acronym.

**p1 l1ff**  That first sentence "Possible future trajectories of the Earth system in the Anthropocene are determined by the increasing entanglement of processes operating in the physical, chemical and biological systems of the planet, as well as in human societies, their cultures and economies" is very debatable. "Possible" is redundant, the choice of Anthropocene (capitalized) possibly politically motivated, the word "determined" raises concern of confusion with "deterministic" approaches and the conjunctions are not well placed. If I may rephrase this, the "Anthropocene (sic!) is characterized by close entanglement between the Earth system and its physical, chemical and biological processes and the World system with its economic, social, and cultural interactions." And certainly there is no need for eight (!) citations to entanglement in the Anthropocene; possibly, authors who argue for entanglement in the anthropocene (minuscule "a") should be cited instead.

**p1 l3ff**  Second sentence "Here, we introduce the copan:CORE open source software library that provides a framework for developing, composing and running World-Earth models..." This sentence should foremost and first emphasize that this publication introduces a new term and concept, namely that of a WEM, and second that it also provides a software library for modeling such WEM. Also the definition of WEM as "social-ecological co-evolution up to planetary scales" does not agree exactly with the later definitions given in the manuscript. Please elaborate in the abstract on your term WEM, on the theoretical embedding and reduce the room
given to technicalities.

**3  Introduction**

**p2 l25ff**  Please provide a reference your historical examples. In the discussion of the "Tragedy of the Commons" it would not hurt to point to related works that make Ostrum's work operational in model simulations.

**p2 l34f**  I believe the term "planetary social-ecological system" needs more explanation. SES are usually understood as local in much of the literature, and as multiple instances that behave very different. Thus, also the implementation of SES mostly in agent-based models (as you mention yourself later in the introduction). Elaborate and contrast your "planetary" approach to the local SES. You might also consider to reduce usage of the term SES altogether in favor of your new term WEM to avoid this confusion.

**p3l 7ff**  Congratulations on the choice of the term "World Earth Model". This is to date the best term I have yet heard to describe the type of model you've developed. I suggest to elaborate on how you come to this term, and to set it off from other terms including, but not limited to, SES and CHANS (Coupled Human and Natural Systems).

**4  Blueprinting World Earth Models**

**p3 l6ff**  Please use precise language, do not "outline guidelines" or "address leading research questions". Check entire manuscript for this type of bloated wording.

**p3 l7ff** For the definition of an Anthropocene you already need to say how it differs from the Holocene and other paleoclimatic stages. So the first half of question type 1 is circular. As for the second part "how might it alter the evolution", it is unclear what "it" refers to. Certainly the "Anthropocene" is not an actor (so it cannot alter) but a diagnostic term for the World-perturbed Earth. Please clarify.

**p3 l8ff** Avoid general valueing statements like "disastrous" or specify; check entire manuscript for further occurences of such type. Avoid jargon here and explain all domain-specific terms.

**p3 l27** Here you use "framework" in the management sense, later you use (software) "framework" for the technical description. Then you both consider Netlogo as well as copan:CORE frameworks, but both are very different things. I would prefer to term NetLogo a modeling platform. The term "framework" is a difficult one, please try to use it consistently in only one sense (and explain that sense by giving your definition of a framework) throughout the paper.

**p3 l27** The "high degree of modularity and flexibility and coupling capabilities" is not substantiated. While there is some software modularity and role modularity (see my later comment), there is no effort made towards coupling capabilities in a more general sense (there is a statement later on interoperability with LPJml, see my comment below). There is also no elaboration of what you mean by flexibility.

**p4 l14ff** I don't see how the stylized biophysical description in the WEM can help answer this question. Would we not need a "whole" WEM where both the Earth System and the Socio-cultural system are described process-detailed (ref your Fig 1)?

**p4 l25ff** You argue that environmental and societal processes should be described on a similar level complexity, yet in Figure 1 you argue for a stylized description of the biophysical world. Please explain better or resolve this conflict between text

and figure. As for your list of five characteristics of WEM, I suggest to give each item a short title. You might want to consult our modeling framework paper (see references, we had to argue for biological models on par with physical oceanography models and called this "equitability"). Others could be "nonlinearity" and "aggregation".

**5 copan:CORE WEM framework**

**p6 l22ff** Your modularity is achieved through object-oriented programming. This is not enough to justify modularity as an eminent feature of your software. This is mere good software practice. Object-oriented programming then does not per se allow interoperability and dynamics coupling to other models, as you claim. To this end, much more (like coupling frameworks, data exchange standards, computational bridging infrastructures) are needed, all of which are absent from the manuscript. Please elaborate on the specific coupling solution to LPJml and to IMAGE to substantiate your interoperability claim.

**p8 l14ff** Consider making this list of process-types identical to the one found in figure 2

**p9 l16ff** It should also be the role of the "master" model to ensure interoperability with other modeling frameworks, of which you make no mention. A prominent framework that you should reach out to is the CSDMS BMI (basic model interface) idea. Your master component could implement that BMI/CMI and thus make all user-contributed models also interoperable. We have, e.g., done this with the FABM (Framework for Adaptive Biogeochemistry) for ESMF interoperability. If you don't want to add a BMI (to CSDMS, OpenMI or ESMF, or other frameworks) please add a section outlining your plans to do so or your reservations against doing so.

**p13 l 3ff** The term "modular" is in your context the software modularity typically found in modern software architecture. This is \*not\* an emanating feature of co-pan:CORE. There is modularity beyond software modules in other frameworks and I would encourage you to rethink modularity in that broader sense.

**p16 l 4** A section on performance is missing (e.g. at end of section 3). Many thousands of cells, individuals or other entities might have to be simulated with this framework. What is your approach to ensuring that integrations of differential equations (exemplary for one of your process-types) is efficiently handled for large numbers of entities? Is there consideration for optimization (you already mention communication with MPI and JSON) for high-performance computing architectures? What tradeoffs to performance do you expect by using "slow" packages like sympy? Did you perform any scaling experiments?

**6  Figures**

Overall, the figures are of mixed quality and style. A more consistent layout, style, coloring and fonts across all figures would make the paper more pleasing to the eye and also more readable. Please spend some efforts towards this goal. Especially Figs 1 and 2 are very clear and could serve as a template.

**fig 1** The white box could contain text, such as "none"

**fig 2** For consistency with text, use "process type", not "modelling approach"

**fig 3** This entity–relationship diagram in UML style is only understandable to a small fraction of readers. Please explain the notation used in the diagram (for example by giving an example of the cell–person relationship). I do not at all understand

the circular relationships for entities with themselves, especially for the SocialSystem entity. Please clarify. This figure does not need color, in fact, color distracts here.

**fig 4** This "spaghetti" diagram is not helpful. Please create an entirely new graph. Rearrange the information, e.g., choose a UML style for consistency with fig 3. Avoid crossing lines, strange coloured shapes without obvious semantics, use typewriter font consistently for code parts. Make graphical markers (colors, line widths, boxes) easily accessible by adding a legend instead of explanation in caption.

**fig 5** see comments for code figures later

**fig 6** Change colours entirely to be consistent with figure 2 (CUL, MET, ENV). Don't use background color. Change layout to something visually appealing; currently the table structure suggest as semantic for rows and columns that is not evident.

**fig 7** Table layout conveys meaning, but could be highlighted (columns are scenario (is that what you call "runs" in the caption?, rows are taxa). Avoid mixing colour semantics with those of previous figures. Avoid mixing color semantics between panels: How to top and middle row colors align? If they do, don't add two legends but use only one. Explain why for CUL/ENV there are only four quantities shown, but for MET there is an ensemble (each four) of three quantities shown. Upper left: where is the blue line (I guess hidden behind the grey one ...)? Find a way to display lines that are on top of each other without hiding any (also upper right figure). Possibly add events on time axis, especially for understanding middle right panel events with sudden transitions from fossils to biomass.

**figs 5,8-10** Try improved syntax colouring and choose different font. Fixed width is important, but better use a smaller width. Consider light grey for comments, for
example. A light (cream) background might help to set the code apart from the title, which is barely visible (and which uses inconsistent font with main text).

**Technical comments**

**p7 l 10** There is no such thing as "sharp criteria". Criteria alone is sufficient.

**p14 l14** The link to pycopancore (http://github.com/pik- 15 copan/pycopancore) does not work yet (so make sure it does work on publication day)

**p14 l29ff and Figure 5** Use a consistent form for presenting code, do not alternate between text and figure.

**p16 l 4** Examplary => Exemplary

**p16 l 9** "not intended to be a serious representation". A representation cannot be serious. I suggest "is intended to be a toy representation". BTW, what is the "real" world anyway :=)

**p17 l3ff** Avoid double parentheses throughout this paragraph and manuscript.

**References**

Lemmen, C., Hofmeister, R., Klingbeil, K., Nasermoaddeli, M. H., Kerimoglu, O., Burchard, H., Kösters, F., and Wirtz, K. W.: Modular System for Shelves and Coasts (MOSSCO v1.0) – a flexible and multi-component framework for coupled coastal ocean ecosystem modelling, Geosci. Model Dev., 11, 915-935, https://doi.org/10.5194/gmd-11-915-2018, 2018.

---

## Author Comment (AC1) · 27 Jul 2018

Earth System Dynamics Discussions Article

**Earth system modelling with complex dynamic human societies: the copan:CORE World-Earth modeling framework**

Final authors' response

Jonathan F. Donges*, Jobst Heitzig*, Wolfram Barfuss, Johannes A. Kassel, Tim Kittel, Jakob J. Kolb, Till Kolster, Finn Müller-Hansen, Ilona M. Otto, Marc Wiedermann, Kilian B. Zimmerer, and Wolfgang Lucht

*(Reviewers' comments cited in italics)*

**Response to Brian J. Dermody (reviewer 1)**

*The authors present a modelling framework for a new generation of Earth System Models that they term World Earth Models (WEM). The paper presents their theoretical framework for capturing environmental, cultural and what they term, social metabolism processes in a linked model. They then provide details of the software package copan:CORE, which builds on their theoretical framework and is implemented in Python language.*

*It is this reviewers opinion a new generation of Earth System Models is urgently needed to capture complex dynamics between humans and the environment and this paper is an important first step in attempting to implement a modelling framework for a WEM.*

We are happy that you share our opinion on the need for a new generation of models and thank you for your overall assessment of our attempt.

*However, I would like to see more argumentation for the development of the theoretical framework they set out as well as clearer description of the model implementation, with consistency between the description of the theoretical framework and the model implementation framework. In addition, I suggest some structural changes to the paper.*

We will attempt to improve the MS in this respect by considering your specific suggestions below. Regarding the consistency between the description of the theoretical framework and the implementation framework, we are not completely sure where you find them inconsistent, so we will check carefully during the revision process that these two levels of description are more easily matched by the reader. We would welcome any further comments on where there might be inconsistencies.

*Paper Structure*

*I think the paper could benefit from a slight restructuring for sections 1–3. The introduction introduces many terms without explanation or explanation comes in section 2 and 3. One important example of this is the term social-metabolism. This is later defined along with the other theoretical framings of the Earth System: environment and culture. You should introduce this framing earlier.*

We realize we might have misjudged the commonality of terms such as "social metabolism", which might, though well-established in some research communities, be unfamiliar to part of ESD's readership. We will make sure to identify such terms by having the MS read again by a more traditional Earth System scientist and will accordingly give their definitions earlier.

*So, I would recommend starting with a shorter introduction with section 1.1 outlining the current state of modelling earth system processes, the shortcomings of these approaches and the motivation for a new framework.*

*Then section 1.2 outline briefly, and in a language that users can follow (so if you introduce a new term such as social-metabolism, explain it), your theoretical framework, what problems it addresses and how it is implemented.*

We agree that such a summary will allow the reader to get a faster general understanding of what will follow, so we are happy to add it as a new section 1.2.

If we understand the following part of your comments correctly, you suggest to either remove the current section 2, which describes the guiding principles we suggest for World-Earth Models, or to include it in much shorter form in 1.1., or to merge it with the description of the theoretical modeling framework (which is currently section 3). Since you comment on parts of this current section 2 below, we assume you would agree that they should not be deleted completely. Because these principles constitute part of the motivation for particular choices we made in designing our framework, we believe that they need to stay included in some way, but may be shortened considerably. To keep the logic of the MS consistent, we believe these principles need to be given before summarizing the framework as you suggested in a new section 1.2. The most natural place for them therefore seems to be the beginning of that new section. So, we will restructure the first sections as follows:

> 1  Introduction
> > 1.1  Motivation (content as you suggested)
> > 1.2  Towards blueprinting World-Earth models (shortened version of current section 2 followed by summary of framework as you suggested)
> 2  The copan:CORE World-Earth modelling framework (currently section 3)

*Then in section 2 outline the theoretical framework in more detail. Crucial here is to motivate your reasoning behind the choices you make. This is not always clear in the discussion manuscript (Theoretical reasoning behind framework).*

We believe that by "theoretical framework" you refer to the software-independent level of description of our framework that currently forms section 3.1, "Abstract structure". We agree that its details might need a better motivation in terms of the reasoning presented in the earlier part of the MS. At the same time, we must make sure that this part can still serve as a concise reference to the main concepts used in our framework that is not cluttered by too much background information and motivation. We will therefore solve this by adding to the end of each subsections of this section a paragraph labelled "Rationale", giving the reasoning you rightly request. So, the beginning of the new section 2 will look like this:

> 2  The copan:CORE World-Earth modelling framework
> > 2.1  Abstract structure
> > > 2.1.1  Entities, processes, attributes
> > > > …
> > > > Rationale: …
> > > 2.1.2  Entity-types, process taxa, process-types
> > > > …
> > > > Rationale: …
> > > …

The software design section (currently 3.4) will thus become 2.4

*Then section 3 outline how the model is implemented. It should be very clear how the theoretical framework links with the implementation framework. This is not currently clear to me yet. Figure 2 is helpful, but I would like to see then how that relates to model framework structure: i.e. a figure like figure 4 but then capturing what is shown in figure 2. Importantly, if you keep more consistency between the theoretical framework structure and the implementation structure, then readers and users will be more easily able to follow what you have done.*

We agree that the current description of our reference implementation of the framework in the Python language (currently 3.5) is less complete than the theoretical description of the modeling framework's concepts (currently 3.1–3.3) and the language-independent description of the software design (currently 3.4), and hence the link between the theoretical concepts and the individual Python features we mention may not be sufficiently clear.

Still, we feel that we should not add much more detail to this lowest-level description of the software for several reasons. On the one hand, the current implementation in Python is mainly meant as a first reference implementation which readers may use to try out the framework but whose details might undergo significant changes and improvements in future releases and will probably be accompanied by more high-performance-oriented alternative implementations of the same framework in other languages, in particular C++ and potentially Julia, so that a detailed description as part of the MS will soon be outdated. On the other hand, more importantly, ESD is not a software journal and we believe that software implementation details are not important for the scientific understanding of the framework, its design and possible merit for scientific research.

In view of this situation and the length of the MS we therefore plan to restructure the MS regarding the implementation description as follows. The current subsection 3.5, "Reference implementation in Python" will become subsection 2.5 but only its current first paragraph and the first code example (Fig. 5) will stay in the main text, extended by a sentence saying that the most recent API documentation can be found online. The rest of the current 3.5 will be moved into the SI, including the current Fig. 4 which we will rework to be more easily accessible and visually more appealing.

To visualize the different elements of the software more clearly, we use the freed space to improve Fig. 3 and add a new Fig. 4 as follows. Fig. 3, currently showing a class diagram only for entity-types, will be completed to show all classes that correspond to the abstract concepts shown in Fig. 2, in an arrangement corresponding to Fig. 2. i.e., we will add the classes "Culture", "Metabolism", and "Environment" to its left and the classes "Step", "Explicit", "ODE", "Implicit", "Event" etc. to its right. The new Fig. 4 will show in a simple way how several model components contribute mixin-classes to the entity type implementation classes of a composed model.

*Theoretical reasoning behind framework*

*Page 4 Line 10: The planetary boundaries concept has come in for some criticism lately (Montoya et al. 2018). A model framework such as this can potentially explain how planetary boundaries emerge through cross-scale human-environment interactions. It would be good to explain shortly how such a model framework could illuminate how we can understand how global planetary boundaries link across scales, keeping in mind the criticisms of the concept.*

We thank the referee for this helpful comment. We will add a more detailed discussion of how World-Earth modeling can help to understand the properties of planetary boundaries as emergent

properties of complex social-ecological systems on the global scale, reflecting also on different perspectives on the planetary boundaries concept.

*Page 4 Line 25 "environmental and societal processes should be described on similar levels of complexity" – sounds good but why? And what does that mean in reality? A tree and a person is equivalent? A country and an ecosystem equivalent? If so, what is the theoretical grounds for that?*

We aim to state here that in our opinion World-Earth models should contain balanced representations of social and biophysical components of the Earth system. They should neither be too biased towards very detailed biophysical processes (as current Earth system model already cover this terrain) or towards very detailed socio-economic processes (as current Integrated Assessment Models [IAMs] cover part of this terrain already). Still, concrete model design needs to follow the requirements of the research questions at hand. In the revised manuscript, we will provide a more differentiated reasoning behind this guideline for WEMs.

*Page 4 line 30–33, page 5 line 0–5 This seems all reasonable but why? And what is your grounding for these statements? In addition, there is a large body of work on applying agent-based models in the socio-cultural domain, which seems to have been ignored here. If you want to capture that, then you should demonstrate that you are aware of this literature and have considered it, including the many pitfalls of applying agent-based models to social systems. Also relates to the statement on Page 6, Line 13–15. There has been extensive work on formal modelling of socio-cultural processes. See Netlogo References for example: https://ccl.northwestern.edu/netlogo/references.shtml*

We will add an explicit explanation of why we think that agent-based (ABM) and network modelling approaches are a valuable addition to Earth system modelling and should, hence, be implementable in World-Earth models. We will emphasize the point that while there is a rich literature on ABMs and formal modelling of socio-cultural processes, it has so far been weakly integrated with other Earth system processes in Earth system modelling. World-Earth models are intended to be designed as tools to enable this integration and coupling that is missing so far.

*Page 5 line 9–10 Outline why it is important to capture tipping points. This should also be covered in the intro when discussing shortcomings of existing models.*

We thank the referee for pointing out yet another missing explicit explanation for one of our suggested guidelines. We will revise the text accordingly, highlighting that a major shortcoming of existing models in the Earth system domain (particularly IAMs) is their inability to represent social-economic or social-ecological or social tipping points.

*Coupling or not?*

*It is not clear to me whether the copan:CORE framework is designed to couple to other models such as LPJ-Guess and IAMs that you mention or if it is a standalone model with different modules or both? I.e. can external models can be modules within the copan:CORE framework? I would encourage you to outline this in more detail and with more prominence in the paper as a lot of the community are interested in a framework for coupling existing models that can incorporate the kind of dynamics you set out to include.*

This is a really important remark which also very much resonates with Mr. Lemmen's comments. We admit we should have discussed the coupling issue in much more detail and will do so in a new subsection 2.1.6, "Interoperability with other model software". To answer your question already here, at the moment it is essentially possible to include external model software by writing a short

"wrapper component" that handles the exchange of state data, including any necessary regridding, and calls the external model's time-stepping function as long as the external model provides some interface that allows this (e.g. by implementing a BMI, see Mr. Lemmen's comments below). For IAMs that run in intertemporal optimization mode rather than time-forward simulation mode (via stepping or integration) this will probably not be possible since copan:CORE currently only supports time-forward simulation mode.

*Model description*

*Generally, I find the model description too vague to know what can and can't be done with it. For instance, it is mentioned you can model resources flows and migration with it. How would this be implemented? Perhaps a few simple examples of specific model frameworks would help the reader understand what copan:CORE can and cant do. E.g. explain how would you use the framework to capture the relation between migration and drought or how tragedy of the commons scenarios emerges within a river catchment?*

We were hoping the exemplary model described in the current Sec. 4 would suffice to answer this question. As described in the SI, it implements some resource flows via ODEs in its carbon cycle and economic production components, and has other ODEs implementing migration in its "wellbeing-driven migration" component. The bottom lines of Fig. 10 show a code example, how ODEs are specified in the Python reference implementation. Regarding the modeling of a possible relationship between migration and drought, a model component developer has many possibilities: she could "micro"-model individual migration decisions by giving her "Individual" entity-type mixin class a process of type "Event" that makes the individual move to a different "SocialSystem" at some regular or random time-points with some probability depending on some attribute of its current "Cell" of residence that represents the occurrence of a drought. Or she might choose a "macro"-modeling approach by giving the "Cell" mixin class a process of type "Explicit" that specifies an explicit equation which computes at each time point the emigration flow from this place as a function of some drought-related cell attributes. For tragedy-of-the-commons scenarios the model component developer might chose a game-theoretic modeling approach and give each "Cell" representing a catchment a process of type "Step" that represents discrete time-points at which all "Individual"'s residing in the cell make water extraction decisions; the outcome of these decisions might by implemented by giving each "Cell" another process of type "Implicit" that encodes a system of implicit equations which represent the Nash equilibrium between these individual decisions. Implicit equations can also be used to model price equlibria. We have chosen the code examples in Figs. 9 and 10 to show all available formal process types.

*Specific Comments*

*Page 1 line 1–3. This is quite a vague opening sentence. I would drop it. Start with: we introduce.… (the abstract is already quite long)*

This first sentence of the abstract was intended to motivate the need for a new class of Earth system models. We will attempt to weave this motivation into the remainder of the abstract and generally shorten the abstract overall.

*Page 1 line 5: Not clear what is meant by user roles. Can you be more explicit, especially since this is the abstract.*

We will give examples which become clearer later: model end user, model composer, model component developer.

*Page 1 Line 14: I wouldn't include social metabolism in the abstract. Not a widely know term.*

We will add a brief explanation of the term or replace it by a more widely known term in the abstract.

*Page 3 line 30–35 Is this an agent-based model? From the abstract and introduction, I thought it was more than that. However, this concluding paragraph makes the reader think that you are going to introduce an ABM. If you view it as an ABM, fine. But then state that clearly in the abstract.*

World-Earth models in our understanding contain agent-based process representation along with other modules that may be, e.g. grid-based. We will clarify this at the indicated place in the text and where else appropriate.

*Page 5 line 8 what is time forward integration?*

We will define it like this: simulation of changes in system state over time consecutively in discrete time-steps (e.g. via difference equations or stochastic events) or at a continuum of time points (e.g. via ordinary differential equations), rather than solving equations that describe the whole time evolution at once as in intertemporal optimization.

*Page 7 Figure 2. While I like this figure, it could be clearer. It's not clear to me how each of the elements relate. Is each level of the network equivalent to Cul, Met and Env and are they then equivalent to the network on the right? It could be simpler to just show the central image (entity types) in one figure in the new intro section 1.2, for example. I like the way you show the different modelling approaches but it isn't clear with the network image how they relate or how cul,met, env relate.*

The current version of the Fig. is the result of a lot of discussions with colleagues. We found it important to make clear that there are these three different aspects of WEMs, process taxa, entity types, and modeling approaches, and that they are loosely related without having a simple one-to-one relationship. The thicker lines between "CUL" and "individuals", "MET" and "social systems", and "ENV" and "cells" indicate that we expect that most socio-cultural processes will be implemented at the level of individuals, most socio-metabolic processes at the level of social systems, and most environmental processes at the level of grid cells. The thinner lines however are meant to make clear that this is by no means necessary and that some socio-cultural processes (e.g. regular elections) might better be implemented at the level of social systems etc. The same holds for the relationship between entity types and modeling approaches. While agent-based model components will probably most often use the "individual" entity type, they might also use the "social system" entity type, e.g. for representing governments' decisions, etc.

*Section 3.1 is very clear.*

*Page 10 line 4-5 delete "maybe changing numbers and"*

Here we disagree since we believe it is a notable feature that during a simulation, the number of, say, individuals may change.

*Page 10 line 12 instead of following entity types, write entity types outlined in Section 3.2*

OK.

*Page 10 line 20: Give an example, such as countries to clarify*

OK.

*Page 11 line 4: human-designed, human-reproduced*

OK.

*Page 12 Line 6–30 Introduce this earlier in the manuscript. See my comments on structure*

We agree and will edit the introduction accordingly, see also our response on restructuring the sections above.

*Page 16 Line 4: Examplary is not a word. It appears to be an obsolete form of exemplary which means "perfect".*

We will correct this typo of "exemplary".

*A couple of important references "in prep". Try to find pre-existing publications to support arguments in addition to these where possible.*

The Donges et al., in prep., paper is now published as a discussion paper in Earth System Dynamics (currently in review). We will update the reference accordingly. Regarding the Otto et al., in prep., paper which is currently in review but not published online, we will support it by already published literature on the topic.

*Page 22: Figure nested within references*

We'll move these to the SI as stated above.

---

## Author Response (AR1)

**Earth System Dynamics Manuscript**

**Earth system modeling with complex dynamic human societies: the copan:CORE World-Earth modeling framework**

Point by point reply

Jonathan F. Donges\*, Jobst Heitzig\*, Wolfram Barfuss, Johannes A. Kassel, Tim Kittel, Jakob J. Kolb, Till Kolster, Finn Müller-Hansen, Ilona M. Otto, Marc Wiedermann, Kilian B. Zimmerer, and Wolfgang Lucht

(Reviewers' comments cited in italics)

**Summary**

Based on our earlier final authors response on the discussion paper, we revised the manuscript thoroughly to address the issues raised, deviating only very slightly from the detailed plan we had outlined in the final authors response. We therefore mainly repeat the point by point reply from the final authors response and indicate in blue where we deviated from it.

**Response to Brian J. Dermody (reviewer 1)**

The authors present a modelling framework for a new generation of Earth System Models that they term World Earth Models (WEM). The paper presents their theoretical framework for capturing environmental, cultural and what they term, social metabolism processes in a linked model. They then provide details of the software package copan:CORE, which builds on their theoretical framework and is implemented in Python language.

It is this reviewers opinion a new generation of Earth System Models is urgently needed to capture complex dynamics between humans and the environment and this paper is an important first step in attempting to implement a modelling framework for a WEM.

We are happy that you share our opinion on the need for a new generation of models and thank you for your overall assessment of our attempt.

However, I would like to see more argumentation for the development of the theoretical framework they set out as well as clearer description of the model implementation, with consistency between the description of the theoretical framework and the model implementation framework. In addition, I suggest some structural changes to the paper.

We improved the MS in this respect by considering your specific suggestions below. Regarding the consistency between the description of the theoretical framework and the implementation framework, we are not completely sure where you find them inconsistent, so we checked carefully during the revision process that these two levels of description are more easily matched by the reader.

**Paper Structure**

I think the paper could benefit from a slight restructuring for sections 1–3. The introduction introduces many terms without explanation or explanation comes in section 2 and 3. One important example of this is the term social-metabolism. This is later defined along with the other theoretical framings of the Earth System: environment and culture. You should introduce this framing earlier.

We realize we might have misjudged the commonality of terms such as "social metabolism", which might, though well-established in some research communities, be unfamiliar to part of ESD's

readership. We made sure to identify such terms by having the MS read again by a more traditional Earth System scientist and have accordingly given their definitions earlier.

So, I would recommend starting with a shorter introduction with section 1.1 outlining the current state of modelling earth system processes, the shortcomings of these approaches and the motivation for a new framework.

Then section 1.2 outline briefly, and in a language that users can follow (so if you introduce a new term such as social-metabolism, explain it), your theoretical framework, what problems it addresses and how it is implemented.

We agree that such a summary will allow the reader to get a faster general understanding of what will follow.

If we understand the following part of your comments correctly, you suggest to either remove the original section 2, which describes the guiding principles we suggest for World-Earth Models, or to include it in much shorter form in 1.1., or to merge it with the description of the theoretical modeling framework (which is originally section 3). Since you comment on parts of this original section 2 below, we assume you would agree that they should not be deleted completely. Because these principles constitute part of the motivation for particular choices we made in designing our framework, we believe that they need to stay included in some way, but may be shortened considerably. After giving this much thought, we have decided to reorder the first sections as follows, giving them significantly more structure than before:

- 1 Introduction and theoretical considerations
  - 1.1 Motivation
    - 1.1.1 State of the art
    - 1.1.2 Current gap in the Earth system modeling landscape

1.1.3 World-Earth modeling: a novel approach to Earth system analysis of the Anthropocene

- 1.1.4 Features of the copan:CORE modeling framework
- 1.2 General characteristics of integrated World-Earth models
  - 1.2.1 Basic process taxa in World-Earth models
  - 1.2.2 Design principles for World-Earth models

1.2.3 World-Earth models compared to existing modeling approaches of global change

2 The copan:CORE World-Earth modeling framework (originally section 3)

Then in section 2 outline the theoretical framework in more detail. Crucial here is to motivate your reasoning behind the choices you make. This is not always clear in the discussion manuscript (Theoretical reasoning behind framework).

We believe that by "theoretical framework" you refer to the software-independent level of description of our framework that originally forms section 3.1, "Abstract structure". We agree that its details might need a better motivation in terms of the reasoning presented in the earlier part of the MS. At the same time, we must make sure that this part can still serve as a concise reference to the main concepts used in our framework that is not cluttered by too much background information and motivation. We have therefore solved this by adding to the end of each subsections of this section a paragraph labelled "Rationale", giving the reasoning you rightly request. So, the beginning of the new section 2 now looks like this:

- 2 The copan:CORE World-Earth modeling framework
  - 2.1 Abstract structure
    - 2.1.1 Entities, processes, attributes
      - Rationale: ...
    - 2.1.2 Entity types, process taxa, process types

Rationale: ...

Since we moved the section on process taxa to the introduction (see above), the software design section (before 3.4) thus became 2.3.

Then section 3 outline how the model is implemented. It should be very clear how the theoretical framework links with the implementation framework. This is not currently clear to me yet. Figure 2 is helpful, but I would like to see then how that relates to model framework structure: i.e. a figure like figure 4 but then capturing what is shown in figure 2. Importantly, if you keep more consistency between the theoretical framework structure and the implementation structure, then readers and users will be more easily able to follow what you have done.

We agree that the original description of our reference implementation of the framework in the Python language (originally 3.5) was less complete than the theoretical description of the modeling framework's concepts (originally 3.1–3.3) and the language-independent description of the software design (originally 3.4), and hence the link between the theoretical concepts and the individual Python features we mention may not be sufficiently clear.

Still, we feel that we should not add much more detail to this lowest-level description of the software for several reasons. On the one hand, the current implementation in Python is mainly meant as a first reference implementation which readers may use to try out the framework but whose details might undergo significant changes and improvements in future releases and will probably be accompanied by more high-performance-oriented alternative implementations of the same framework in other languages, in particular C++ and potentially Julia, so that a detailed description as part of the MS will soon be outdated. On the other hand, more importantly, ESD is not a software journal and we believe that software implementation details are not important for the scientific understanding of the framework, its design and possible merit for scientific research.

In view of this situation and the length of the MS we therefore restructured the MS regarding the implementation description as follows. The original subsection 3.5, "Reference implementation in Python" became subsection 2.4 but only its original first paragraph and the first code example (Fig. 5) stayed in the main text, extended by a sentence saying that the most recent API documentation can be found online. The rest of the original 3.5 has been moved into the SI, the original Fig. 4 has been dropped because it conveyed no valuable additional information to the new Figs. 3 and 4.

To visualize the different elements of the software more clearly, we uses the freed space to improve Fig. 3 and added a new Fig. 4 as follows. Fig. 3, originally showing a class diagram only for entity-types, was completed to show all classes that correspond to the abstract concepts shown in Fig. 2, in an arrangement corresponding to Fig. 2. i.e., we added the classes "Culture", "Metabolism", and "Environment" to its left and the classes "Step", "Explicit", "ODE", "Implicit", "Event" etc. to its right. The new Fig. 4 shows in a simple way how several model components contribute mixin-classes to the entity type implementation classes of a composed model.

**Theoretical reasoning behind framework**

Page 4 Line 10: The planetary boundaries concept has come in for some criticism lately (Montoya et al. 2018). A model framework such as this can potentially explain how planetary boundaries emerge through cross-scale human-environment interactions. It would be good to explain shortly how such a model framework could illuminate how we can understand how global planetary boundaries link across scales, keeping in mind the criticisms of the concept.

We thank the referee for this helpful comment. We added a more detailed discussion of how World-Earth modeling can help to understand the properties of planetary boundaries as emergent properties of complex social-ecological systems on the global scale, reflecting also on different perspectives on the planetary boundaries concept. Page 4 Line 25 "environmental and societal processes should be described on similar levels of complexity" – sounds good but why? And what does that mean in reality? A tree and a person is equivalent? A country and an ecosystem equivalent? If so, what is the theoretical grounds for that?

We aim to state here that in our opinion World-Earth models should contain balanced representations of social and biophysical components of the Earth system. They should neither be too biased towards very detailed biophysical processes (as current Earth system models already cover this terrain) or towards very detailed socio-economic processes (as current Integrated Assessment Models [IAMs] cover part of this terrain already). Still, concrete model design needs to follow the requirements of the research questions at hand. In the revised manuscript, we provided a more differentiated reasoning behind this guideline for WEMs.

Page 4 line 30–33, page 5 line 0–5 This seems all reasonable but why? And what is your grounding for these statements? In addition, there is a large body of work on applying agent-based models in the socio-cultural domain, which seems to have been ignored here. If you want to capture that, then you should demonstrate that you are aware of this literature and have considered it, including the many pitfalls of applying agent-based models to social systems. Also relates to the statement on Page 6, Line 13–15. There has been extensive work on formal modelling of socio-cultural processes. See Netlogo References for example: https://ccl.northwestern.edu/netlogo/references.shtml

We added an explicit explanation of why we think that agent-based (ABM) and network modelling approaches are a valuable addition to Earth system modelling and should, hence, be implementable in World-Earth models. We emphasized the point that while there is a rich literature on ABMs and formal modelling of socio-cultural processes, it has so far been weakly integrated with other Earth system processes in Earth system modelling. World-Earth models are intended to be designed as tools to enable this integration and coupling that is missing so far.

**Page* 5 *line* 9–10 *Outline why it is important to capture tipping points. This should also be covered in the intro when discussing shortcomings of existing models.**

We thank the referee for pointing out yet another missing explicit explanation for one of our suggested guidelines. We revised the text accordingly, highlighting that a major shortcoming of existing models in the Earth system domain (particularly IAMs) is their inability to represent social-economic or social-ecological or social tipping points.

**Coupling or not?**

It is not clear to me whether the copan:CORE framework is designed to couple to other models such as LPJ-Guess and IAMs that you mention or if it is a standalone model with different modules or both? I.e. can external models can be modules within the copan:CORE framework? I would encourage you to outline this in more detail and with more prominence in the paper as a lot of the community are interested in a framework for coupling existing models that can incorporate the kind of dynamics you set out to include.

This is a really important remark which also very much resonates with Mr. Lemmen's comments. We admit we should have discussed the coupling issue in much more detail and will do so in a new subsection 2.3.5, "Interoperability with other model software". To answer your question already here, at the moment it is essentially possible to include external model software by writing a short "wrapper component" that handles the exchange of state data, including any necessary regridding, and calls the external model's time-stepping function as long as the external model provides some interface that allows this (e.g. by implementing a BMI, see Mr. Lemmen's comments below). For IAMs that run in intertemporal optimization mode rather than time-forward simulation mode (via stepping or integration) this will probably not be possible since copan:CORE currently only supports time-forward simulation mode.

**Model description**

Generally, I find the model description too vague to know what can and can't be done with it. For instance, it is mentioned you can model resources flows and migration with it. How would this be implemented? Perhaps a few simple examples of specific model frameworks would help the reader understand what copan: CORE can and cant do. E.g. explain how would you use the framework to capture the relation between migration and drought or how tragedy of the commons scenarios emerges within a river catchment?

We were hoping the exemplary model described in the original Sec. 4 would suffice to answer this question. As described in the SI, it implements some resource flows via ODEs in its carbon cycle and economic production components, and has other ODEs implementing migration in its "wellbeing-driven migration" component. The bottom lines of the original Fig. 10 (now moved to the SI for space reasons) show a code example, how ODEs are specified in the Python reference implementation. Regarding the modeling of a possible relationship between migration and drought, a model component developer has many possibilities: she could "micro"-model individual migration decisions by giving her "Individual" entity-type mixin class a process of type "Event" that makes the individual move to a different "SocialSystem" at some regular or random time-points with some probability depending on some attribute of its current "Cell" of residence that represents the occurrence of a drought. Or she might choose a "macro"-modeling approach by giving the "Cell" mixin class a process of type "Explicit" that specifies an explicit equation which computes at each time point the emigration flow from this place as a function of some drought-related cell attributes. For tragedy-of-the-commons scenarios the model component developer might chose a game-theoretic modeling approach and give each "Cell" representing a catchment a process of type "Step" that represents discrete time-points at which all "Individual"s residing in the cell make water extraction decisions; the outcome of these decisions might by implemented by giving each "Cell" another process of type "Implicit" that encodes a system of implicit equations which represent the Nash equilibrium between these individual decisions. Implicit equations can also be used to model price equilbria. We have chosen the code examples in original Figs. 9 and 10 to show all available formal process types.

**Specific Comments**

Page 1 line 1–3. This is quite a vague opening sentence. I would drop it. Start with: we introduce.... (the abstract is already quite long)

We revised the opening sentence to give a clearer motivation of the paper, but decided to keep a motivating sentence in the beginning as a central part of what constitutes an abstract.

*Page 1 line 5: Not clear what is meant by user roles. Can you be more explicit, especially since this is the abstract.*

In shortening the abstract, we dropped the mentioning of user roles from it.

Page 1 Line 14: I wouldn't include social metabolism in the abstract. Not a widely know term.

In shortening the abstract, we also dropped the mentioning of social metabolism from it.

Page 3 line 30–35 Is this an agent-based model? From the abstract and introduction, I thought it was more than that. However, this concluding paragraph makes the reader think that you are going to introduce an ABM. If you view it as an ABM, fine. But then state that clearly in the abstract.

World-Earth models in our understanding contain agent-based process representation along with other modules that may be, e.g. grid-based. We now clarify this in the text.

Page 5 line 8 what is time forward integration?

We now define it like this: simulation of changes in system state over time consecutively in discrete time-steps (e.g. via difference equations or stochastic events) or at a continuum of time points (e.g. via ordinary differential equations), rather than solving equations that describe the whole time evolution at once as in intertemporal optimization.

Page 7 Figure 2. While I like this figure, it could be clearer. It's not clear to me how each of the elements relate. Is each level of the network equivalent to Cul, Met and Env and are they then equivalent to the network on the right? It could be simpler to just show the central image (entity types) in one figure in the new intro section 1.2, for example. I like the way you show the different modelling approaches but it isn't clear with the network image how they relate or how cul, met, env relate.

The original version of the Fig. is the result of a lot of discussions with colleagues. We found it important to make clear that there are these three different aspects of WEMs, process taxa, entity types, and modeling approaches, and that they are loosely related without having a simple one-to-one relationship. The thicker lines between "CUL" and "individuals", "MET" and "social systems", and "ENV" and "cells" indicate that we expect that most socio-cultural processes will be implemented at the level of individuals, most socio-metabolic processes at the level of social systems, and most environmental processes at the level of grid cells. The thinner lines however are meant to make clear that this is by no means necessary and that some socio-cultural processes (e.g. regular elections) might better be implemented at the level of social systems etc. The same holds for the relationship between entity types and modeling approaches. While agent-based model components will probably most often use the "individual" entity type, they might also use the "social system" entity type, e.g. for representing governments' decisions, etc. We now make this clearer in the caption and have reduced the grey-level of those lines to be less prominent and not distract from the main point of the figure.

Section 3.1 is very clear.

Page 10 line 4-5 delete "maybe changing numbers and"

Here we disagree since we believe it is a notable feature that during a simulation, the number of, say, individuals may change.

Page 10 line 12 instead of following entity types, write entity types outlined in Section 3.2

OK.

Page 10 line 20: Give an example, such as countries to clarify

OK.

Page 11 line 4: human-designed, human-reproduced

OK.

Page 12 Line 6–30 Introduce this earlier in the manuscript. See my comments on structure

We agree and edited the introduction accordingly, see also our response on restructuring the sections above.

Page 16 Line 4: Examplary is not a word. It appears to be an obsolete form of exemplary which means "perfect".

We now use "example model" instead.

**A couple of important references "in prep". Try to find pre-existing publications to support arguments in addition to these where possible.**

The Donges et al., in prep., paper is now published as a discussion paper in Earth System Dynamics Discussions (currently in review, 2018). We updated the reference accordingly. Regarding the Otto et al., in prep., paper which is currently in review but not published online, we supported it by already published literature on the topic.

**Page 22: Figure nested within references**

We moved these to the SI as stated above.

**Response to Carsten Lemmen (reviewer 2)**

**1 General comments**

This manuscript by Donges and colleagues introduces the core technology and concept behind a new software tool called "copan", that should serve as "a framework for developing, composing and running World-Earth models". The authors motivate the development of such World Earth Models (WEM) that encompass dynamic descriptions of both the anthroposphere as well as the Earth System, they contrast WEM to integrated assessment and Earth System models, they describe the concepts of the developed software package pycopancore and they show simple example applications of the software.

The contribution is within the scope of the Special Issue "Social dynamics and planetary boundaries in Earth system..." in Earth System Dynamics, although the preferred outlet for this kind of technical model description could also be Geoscientific Model Development. The novelty of the approach is the complexity of a World model combined with a stylized version of an Earth model; the innovation is in the open framework and theoretical embedding of the World Earth Model approach.

The paper is overall well written, but suffers from resilience theory and technical jargon, which should be reduced to address a wider readership.

We thank you for this overall assessment and aimed at making the MS more accessible by reducing jargon, especially in the introduction, which has now been shortened in response to Mr. Dermody's comments, and by giving additional definitions where necessary.

Figures are appropriate but they are of mixed graphical quality and accessibility and should be improved on. Tables are appropriate throughout; code examples examples are useful but in need of better quality. The supplementary material is well presented and useful.

In the revision, we aimed at improving the (old and new) figures' and code examples' appearance.

The theory-laden motivation somewhat contrasts with the very technical model description. Reviewer one already remarked on the need for better embedding of these two major perspectives the manuscript assumes. I agree with that assessment, but for brevity I will concentrate in my detailed review below on other aspects of the manuscript. A major missing part is a description of how the presented copan:CORE framework fits into and operates with much of the existing coupling and model infrastructures in Earth and Social sciences; claims to interoperability, modularity and flexibility remain unsubstantiated.

We realize that this had to be improved, see our responses below.

*I* recommend that this paper is published after substantial revisions.

We thank the referee for his overall encouraging assessment.

Title, Abstract and related parts of Introduction

Title. There is an inconsistency in the spelling of "modelling" right in the title. Also, consider to spell out WEM as World Earth Model without hyphens; carefully consider lowercase/uppercase for "Model" in WEM. Nowhere in the paper the authors motivate the naming "copan:CORE"; please add a sentence on this naming and add to a table of abbreviations, if any of this is an acronym.

Thank you for pointing this out. We checked all our spelling again carefully. The hyphen in "World-Earth" has become somewhat of a standard spelling, so after reconsidering it, we decided to keep it. We explain the naming "copan:CORE"; "copan" is the name of PIK's flagship activity for studying coevolutionary pathways, all our models are named "copan:XYZ", and "CORE" refers to the modeling framework which will form the core of our working group's model portfolio.

p1 11ff. That first sentence "Possible future trajectories of the Earth system in the Anthropocene are determined by the increasing entanglement of processes operating in the physical, chemical and biological systems of the planet, as well as in human societies, their cultures and economies" is very debatable. "Possible" is redundant, the choice of Anthropocene (capitalized) possibly politically motivated, the word "determined" raises concern of confusion with "deterministic" approaches and the conjunctions are not well placed. If I may rephrase this, the "Anthropocene (sic!) is characterized by close entanglement between the Earth system and its physical, chemical and biological processes and the World system with its economic, social, and cultural interactions." And certainly there is no need for eight (!) citations to entanglement in the Anthropocene (minuscule "a") should be cited instead.

p1 l3ff. Second sentence "Here, we introduce the copan:CORE open source software library that provides a framework for developing, composing and running World-Earth models..." This sentence should foremost and first emphasize that this publication introduces a new term and concept, namely that of a WEM, and second that it also provides a software library for modeling such WEM. Also the definition of WEM as "social-ecological co-evolution up to planetary scales" does not agree exactly with the later definitions given in the manuscript. Please elaborate in the abstract on your term WEM, on the theoretical embedding and reduce the room given to technicalities.

We thank the referee for these insightful remarks and carefully revised and shortened the abstract accordingly.

**Introduction.**

p2 l25ff Please provide a reference your historical examples. In the discussion of the "Tragedy of the Commons" it would not hurt to point to related works that make Ostrum's work operational in model simulations.

We added such references on historical examples and modelling studies operationalising Ostrom's framework.

p2 l34f I believe the term "planetary social-ecological system" needs more explanation. SES are usually understood as local in much of the literature, and as multiple instances that behave very different. Thus, also the implementation of SES mostly in agent-based models (as you mention yourself later in the introduction). Elaborate and contrast your "planetary" approach to the local SES. You might also consider to reduce usage of the term SES altogether in favor of your new term WEM to avoid this confusion.

p3l 7ff Congratulations on the choice of the term "World Earth Model". This is to date the best term I have yet heard to describe the type of model you've developed. I suggest to elaborate on how you come to this term, and to set it off from other terms including, but not limited to, SES and CHANS (Coupled Human and Natural Systems).

We revised the introduction to define and explain these terms and their interrelations and differences, while making sure that such elaborations do not take too much attention away from the actual aims of the paper.

Blueprinting World Earth Models

p3 l6ff Please use precise language, do not "outline guidelines" or "address leading research questions". Check entire manuscript for this type of bloated wording.

We revised the text to ensure a more concise and crisp language.

p3 l7ff For the definition of an Anthropocene you already need to say how it differs from the Holocene and other paleoclimatic stages. So the first half of question type 1 is circular. As for the second part "how might it alter the evolution", it is unclear what "it" refers to. Certainly the "Anthropocene" is not an actor (so it cannot alter) but a diagnostic term for the World-perturbed Earth. Please clarify.

Well spotted. We carefully revised and clarified this sentence and others relating to the notion of the Anthropocene.

p3 l8ff Avoid general valueing statements like "disastrous" or specify; check entire manuscript for further occurences of such type. Avoid jargon here and explain all domain-specific terms.

We very much agree and revised the text accordingly to avoid unnecessary jargon.

p3 l27 Here you use "framework" in the management sense, later you use (software) "framework" for the technical description.

You are right, this was an unsensible choice of term here. We rephrased this sentence, avoiding the word "framework". We now reserve the word "framework" for its software meaning in the MS.

Then you both consider Netlogo as well as copan:CORE frameworks, but both are very different things. I would prefer to term NetLogo a modeling platform.

We agree since NetLogo provides a graphical interface and other features typical of a modeling platform.

The term "framework" is a difficult one, please try to use it consistently in only one sense (and explain that sense by giving your definition of a framework) throughout the paper.

We added a short definition of "modeling framework" similar to the one of "software framework" that can, e.g., be found on Wikipedia.

p3 l27 The "high degree of modularity and flexibility and coupling capabilities" is not substantiated. While there is some software modularity and role modularity (see my later comment), there is no effort made towards coupling capabilities in a more general sense (there is a statement later on interoperability with LPJml, see my comment below). There is also no elaboration of what you mean by flexibility.

We agree that our discussion of these aspects needs to be improved. We added text to support flexibility in both the introduction and the section on software design; by "flexibility" we mainly mean the possibility to use various combinations of modeling approaches and levels of aggregation (i.e., on the individual, cell, social system, or global level), so that one might combine an ABM of a labour market at the micro-level (i.e., individuals) with a system of ODEs modeling a carbon cycle on the cell level (photosynthesis) and global level (ocean-athmosphere diffusion) and a system of

implicit and explicit equations representing a multisector economy with perfect factor markets on the social system (e.g. country) level.

**p4 l14ff I don't see how the stylized biophysical description in the WEM can help answer this question. Would we not need a "whole" WEM where both the Earth System and the Socio-cultural system are described process-detailed (ref your Fig 1)?**

The simple example WEM described in original Sec. 5 is not meant to be a candidate for a meaningful WEM that could be used to answer real research questions. It is given only to illustrate the features of the modeling framework that this MS is about. If a user of copan:CORE deems it necessary to represent certain processes in more detail than others to be able to answer some specific research question, she can develop a model component that does just that or that acts as a wrapper around an existing external model software implementing these processes (see our comments on coupling in the response to Mr. Dermody and below). Although this is not too relevant here, we however personally believe that the specific question we gave as an example of a research question, namely "How does climate change feed back on complex social structures and their dynamics?", *can* be studied to some extent by a model that has only a stylized biosphere. E.g., changes in global mean temperature can lead to economic damages and increased average mortality, which in turn can lead to changes in demographic structure and economic processes and eventually to changes in social coherence. This is not saying that we already have all the necessary model components or even the theoretical or empirical means to formulate these model components, but that if one had these then a stylized biosphere component might well suffice to perform useful studies.

p4 l25ff You argue that environmental and societal processes should be described on a similar level complexity, yet in Figure 1 you argue for a stylized description of the biophysical world. Please explain better or resolve this conflict between text and figure.

This is a valid point, we carefully revised the text in original Section 2 and Figure 1 to resolve this apparent inconsistency.

As for your list of five characteristics of WEM, I suggest to give each item a short title. You might want to consult our modeling framework paper (see references, we had to argue for biological models on par with physical oceanography models and called this "equitability"). Others could be "nonlinearity" and "aggregation".

We agree with this suggestion and added summary titles to the WEM characteristics, referencing also to your recent modeling framework paper in the same special issue.

**copan:CORE WEM framework**

p6 l22ff Your modularity is achieved through object-oriented programming. This is not enough to justify modularity as an eminent feature of your software. This is mere good software practice. Object-oriented programming then does not per se allow interoperability and dynamics coupling to other models, as you claim.

We believe this is a misunderstanding caused by sloppy wording in the original MS. Of course we do not claim that object-oriented programming automatically leads to either modularity or interoperability. We made sure in the revision that it becomes clearer that the high degree of modularity is the result of very specific design choices (which we found to be easier by following an object-oriented software design pattern rather than, e.g., a functional programming one), such as using multiple inheritance to allow different model components to use the same entities and attributes.

To this end, much more (like coupling frameworks, data exchange standards, computational bridging infrastructures) are needed, all of which are absent from the manuscript. Please elaborate

on the specific coupling solution to LPJml and to IMAGE to substantiate your interoperability claim.

As already hinted at in our response to Mr. Dermody, interoperability with LPJmL, IMAGE etc. follows from the flexibility to basically use any Python code whatsoever in a model components' process implementation methods, including any calls to external software in order to exchange data or call stepper functions etc.

**p8 l14ff Consider making this list of process-types identical to the one found in figure 2**

Perhaps another misunderstanding. There is no list of process types in Fig. 2 but a list of modeling approaches. While there are some one-to-one relationships between the latter and the former, e.g. the modeling approach of using ODEs is supported by providing a process type "ODE" implementing a system of ODEs, other modeling approaches will require several formal process types, e.g. the ABM and adaptive network approaches will typically require a combination of processes of the formal process types "event", "step", and "explicit". We included a similar clarification into the revised manuscript.

**p9 l16ff It should also be the role of the "master" model to ensure interoperability with other modeling frameworks, of which you make no mention.**

We agree that both the "*base* model component" (implementing the most basic entity-types and relationships every copan:CORE model *must* have) and the "master *data* model" (a repository of entity-types and attributes model component developers *may* use) should aim at supporting as much interoperability with other models as possible. copan:CORE's metadata model already contains fields for referencing entries in common variable catalogues such as the *CF Conventions Standard Names* for climate-related quantities or the *World Bank's CETS* list of socio-economic indicators. We realize this should be extended by fields for referencing, e.g., the *CSDMS Standard Names*. We will check whether we missed any further important catalogues and add them if required.

A prominent framework that you should reach out to is the CSDMS BMI (basic model interface) idea. Your master component could implement that BMI/CMI and thus make all user-contributed models also interoperable. We have, e.g., done this with the FABM (Framework for Adaptive Biogeochemistry) for ESMF interoperability. If you don't want to add a BMI (to CSDMS, OpenMI or ESMF, or other frameworks) please add a section outlining your plans to do so or your reservations against doing so.

This is a really very helpful hint, indeed we were sadly unaware of the existence of this initiative. In the next release of pycopancore we will aim at providing a generic wrapper component that allows wrapping external models that implement the BMI into copan:CORE model components, and will also think of how to implement the BMI ourselves in the base model component so that any copan:CORE model can run in a "passive" mode governed by an external coupler that calls its BMI. We added a corresponding paragraph in the revised MS.

p13 l 3ff The term "modular" is in your context the software modularity typically found in modern software architecture. This is \*not\* an emanating feature of copan:CORE. There is modularity beyond software modules in other frameworks and I would encourage you to rethink modularity in that broader sense.

What we mean by modularity in the MS is (i) the division of the program *code* into packages representing "model components" that can be developed by independent "model component developers" and still use the same set of entity-types and attributes, "models" that can be composed from these components by "model composers", and "scripts" that "model end users" can use to perform specific studies, and (ii) the division of each entity-type's processes into contributions coming from different model components via multiple inheritance. We will try to identify further forms of modularity that copan:CORE does or should provide.

p16 l 4 A section on performance is missing (e.g. at end of section 3). Many thousands of cells, individuals or other entities might have to be simulated with this framework. What is your approach to ensuring that integrations of differential equations (exemplary for one of your process-types) is efficiently handled for large numbers of entities? Is there consideration for optimization (you already mention communication with MPI and JSON) for high-performance computing architectures? What tradeoffs to performance do you expect by using "slow" packages like sympy? Did you perform any scaling experiments?

We totally agree that performance is eventually a very important aspect for the production version of any software. With the current paper, the copan:CORE framework described therein, and its reference implementation in Python, pycopancore, our main aim is however a slightly different one than providing a performance-optimized production software. Such a performance-oriented production implementation of the copan:CORE framework, cppcopancore, is currently under development and its performance will be tested and documented thoroughly in a separate paper. For the revision – also in the interest of space – we therefore limited our comments on performance to a sentence stating this and giving running times for the illustrative example.

**Figures**

Overall, the figures are of mixed quality and style. A more consistent layout, style, coloring and fonts across all figures would make the paper more pleasing to the eye and also more readable. Please spend some efforts towards this goal. Especially Figs 1 and 2 are very clear and could serve as a template.

We agree and worked on the figures to achieve a larger degree of stylistic consistency and aesthetics, taking Fig. 1 and 2 as prototypes.

fig 1 The white box could contain text, such as "none"

OK. We added this "none" to clarify the figure.

fig 2 For consistency with text, use "process type", not "modelling approach"

Please see our response above.

fig 3 This entity–relationship diagram in UML style is only understandable to a small fraction of readers. Please explain the notation used in the diagram (for example by giving an example of the cell–person relationship). I do not at all understand the circular relationships for entities with themselves, especially for the SocialSystem entity. Please clarify. This figure does not need color, in fact, color distracts here.

**OK.** Still, because we extended the figure to have a clear correspondence to Fig. 2, we kept some background shading colors to make this clear.

fig 4 This "spaghetti" diagram is not helpful. Please create an entirely new graph. Rearrange the information, e.g., choose a UML style for consistency with fig 3. Avoid crossing lines, strange coloured shapes without obvious semantics, use typewriter font consistently for code parts. Make graphical markers (colors, line widths, boxes) easily accessible by adding a legend instead of explanation in caption.

OK, see also our response to Mr. Dermody.

fig 5 see comments for code figures later

OK.

fig 6 Change colours entirely to be consistent with figure 2 (CUL, MET, ENV). Don't use background color. Change layout to something visually appealing; currently the table structure suggest as semantic for rows and columns that is not evident.

OK. As suggested, we removed background color except the three colors that reference to *CUL*, *MET*, *ENV*. All other background color was replaced by hashing so that different entity types can still be distinguished.

fig 7 Table layout conveys meaning, but could be highlighted (columns are scenario (is that what you call "runs" in the caption?, rows are taxa). Avoid mixing colour semantics with those of previous figures. Avoid mixing color semantics between panels: How to top and middle row colors align? If they do, don't add two legends but use only one. Explain why for CUL/ENV there are only four quantities shown, but for MET there is an ensemble (each four) of three quantities shown. Upper left: where is the blue line (I guess hidden behind the grey one ...)? Find a way to display lines that are on top of each other without hiding any (also upper right figure). Possibly add events on time axis, especially for understanding middle right panel events with sudden transitions from fossils to biomass.

We tried a version in which all lines in the two top panels used the color we used earlier for the CUL taxon, and similarly for the other panels, distinguishing different variables within the same panel by different dash patterns. However, that did not work in the middle right panel since the dash patterns removed too many features of the ragged lines and were not well distinguishable. We therefore retained the original color scheme and instead explained its rationale in the text.

figs 5,8-10 Try improved syntax colouring and choose different font. Fixed width is important, but better use a smaller width. Consider light grey for comments, for example. A light (cream) background might help to set the code apart from the title, which is barely visible (and which uses inconsistent font with main text).

OK.

Technical comments

*p7* l 10 There is no such thing as "sharp criteria". Criteria alone is sufficient.

Although we do not agree here, we removed the "sharp" anyway.

p14 l14 The link to pycopancore (http://github.com/pik- 15 copan/pycopancore) does not work yet (so make sure it does work on publication day)

OK. The following link now works since May 2018: https://github.com/pik-copan/pycopancore . The online publication as open access code was delayed due to institutional legal checks that were pending.

p14 l29ff and Figure 5 Use a consistent form for presenting code, do not alternate between text and figure.

OK.

p16 l 4 Examplary => Exemplary

OK.

p16 l 9 "not intended to be a serious representation". A representation cannot be serious. I suggest "is intended to be a toy representation". BTW, what is the "real" world anyway :=)

OK.

p17 l3ff Avoid double parentheses throughout this paragraph and manuscript.

OK.

**Earth system modelling modeling with complex dynamic human societies: the copan:CORE World-Earth modeling framework**

Jonathan F. Donges1,2,\*, Jobst Heitzig1,\*, Wolfram Barfuss1,3, Johannes A. Kassel1,4, Tim Kittel1,3, Jakob J. Kolb1,3, Till Kolster1,3, Finn Müller-Hansen1,3, Ilona M. Otto1, Marc Wiedermann1,3, Kilian B. Zimmerer1,5, and Wolfgang Lucht1,6,7

[revised manuscript text omitted]

---

## Referee Report (RR1)

Reviewer 1 comments on revised manuscript copan:CORE

I am pleased to comment favourably on this revised manuscript. The authors have made considerable efforts to improve the manuscript. The revised manuscript now clearly outlines the motivation for a new generation of WEMs, the theoretical foundation for their framework and how they have implemented it.

I hope that the authors will be able to successfully develop what they present here. That undertaking will make an urgent and important contribution to understanding the complex drivers of social and environmental change in our globalised world.

**Specific comments**

The revised structure for sections 1.1 and 1.2 is much improved and clearly outlines the current state of modelling earth system processes, the shortcomings of these approaches and the motivation for the new framework presented here as well a helpful introductory overview to the theories and concepts relevant for WEMs.

The addition of a rationale section is a great help for helping the reader understand the motivation and relevance of model design choices.

I think it is a good decision to move most of the detailed code description to the SI and point readers to the most recent API documentation online. I agree with the editor on this that the discussion manuscript was somewhat caught between framework specification which might be more appropriate for a software journal such as Geoscientific Model Development and an introduction and academic motivation of your modelling framework, which I think is what you want to achieve here.

The link between abstract concepts of culture, social-metabolism and environment in figure 2 with figures 3 and 4 is a big help in linking there theoretical grounding for your framework with how it will actually be implemented.

Section 1.2.2 on design principles is much improved and provides a powerful motivation for a new generation of WEMs.

The new section 2.3.5 is a welcome addition, which clarifies how the model may serve to couple internal and external model components.

Page 9, line 16: Perhaps provide the link to the git here.

---

## Author Response (AR2)

**Proposal for a 2nd revision of the copan:CORE paper for Earth System Dynamics**

**J.F. Donges, J. Heitzig and co-authors**

We thank the editor and the referees for their insightful comments that helped to further improve and streamline the presentation of the copan:CORE World-Earth modeling framework in our manuscript under consideration for publication in Earth System Dynamics.

After consultation with the editor James Dyke, we present here a summary of the changes we proposed and implemented for a second revision of our copan:CORE paper under consideration at Earth System Dynamics (ESD). We here mainly build upon the editor's report (see below) that summarized the latest reports by two referees (see also below) and refer to the three main points C1–C3 that our paper intends to make:

> C1 - Argue that coupling of human actions on the Earth systems is necessary or important for a class of scientific and policy-relevant questions.
>
> C2 - Assume C1 (coupling is required) and then demonstrate how such coupling can be implemented into a scalable framework.
>
> C3 - Provide worked examples of the output of a coupling framework that provides new scientific insights and findings that may be of policy relevance.

The following summary of changes takes the role of a response to the editor's comments. Due to the detailed consultation process and since our summary of changes builds on the referees comments, the point-to-point response to their reviews is kept rather brief. (Editor's and referees' comments are in italics below)

Along these lines, we performed the following revisions:

**1. Strengthen C1 in line with James Dykes' editorial summary and Axel Kleidon's review:**

- We now even more clearly identify a set of research questions that need a 'LOOPS' approach and 'World-Earth modelling', and we more clearly define the dividing line to questions where this may not be needed.

- By turning down C2 (see below), we emphasize more how the paper fits into the scope of ESD as a journal and of the Special Issue in particular. The paper is explicitly part of the Special Issue on "Social dynamics and planetary boundaries in Earth system modeling" and we see a valuable role of it in this context, completing that Special Issue's scope ranging from motivational and case-making work via theoretical and methodological considerations towards steps towards practical solutions and case studies.

- Hence, in our perspective the presented copan:CORE framework fits the scope of ESD as an interdisciplinary journal very well (specific points that we address in our paper are highlighted in bold):

  "*Earth System Dynamics (ESD) is a not-for-profit international scientific journal dedicated to the publication and public discussion of studies that take an **interdisciplinary perspective of the functioning of the whole Earth system and global change**. The overall behaviour of the Earth system is **strongly shaped by the interactions among its various component systems**, such as the atmosphere, cryosphere, hydrosphere, oceans, pedosphere, lithosphere, and the inner Earth, but also by life and **human activity**. ESD **solicits contributions that investigate these various interactions and the underlying mechanisms**, **ways how these can be conceptualized, modelled**, and quantified, predictions of the overall system behaviour to global changes, and the **impacts for its habitability, humanity, and future Earth system management by human decision making**.*" (Source: ESD website)

- We push an 'open framework' idea: this addresses the need to be epistemologically flexible because of the large diversity of theories and methodologies from diverse fields in devising "models of man".

- We clarify that our framework can be used to study simple AND complex models, it is explicitly useful for both. (We actually focus on simple models in the copan collaboration most of the time and for good reasons, mainly the same ones that Axel Kleidon and James Dyke mention in their reviews).

- We make sure not to emphasize or promote agent-based modelling, as our references to them were apparently misleading to think that we explicitly want to promote them. We remain agnostic regarding their use, they can be useful for some research questions, less so for others. We now discuss more explicitly and in some detail in the paper when they could be useful, where too simple equation-based models fail: where there is a lot of heterogeneity, where representation of

complex and hierarchical social structures is important, where agency, policies, governance on the level of agents, institutions, social structures is part of the research questions etc. ...

**2. Turn down C2:**

- We now focus on copan:CORE as an open framework for conceptualizing and constructing models of the Earth system in the Anthropocene and the underlying ontology / taxonomy here (with refs to taxonomy paper), independent of software implementation.

- We explain in more detail why such an open framework is useful.

- We moved the detailed description of the software design and implementation to the supplementary information (SI) and describe it only in summary in the main text.

- The focus of this section is, hence, the presentation of the open framework for World-Earth modeling which, in our view, fits well with the scope of the journal Earth System Dynamics.

**3. Strengthen C3:**

In much more detail, we now motivate present and analyse a much simplified version of the exemplary World-Earth model that was discussed in the paper before:

- To improve our proof of concept, we will replace the current section "Example of a World-Earth model implemented using copan:CORE" by a section "Influence of social dynamics in a minimum-complexity World-Earth model implemented using copan:CORE" in which we analyse a reduced version of the current example model, giving all necessary details in the main text and some additional information and possible extensions in the SI.

- The reduced model can be interpreted as a nested box model. On the coarsest level, it has one "planet" box with a maritime and an atmospheric carbon stock. On the middle level, it has two "social systems", representing a "global North" that holds the larger part of capital, and a "global South" that holds the larger part of population. The only social-metabolic (MET) processes we keep are the extraction of fossil fuels, harvesting of biomass, production of renewable energy, production of a final consumption good, and investment into capital growth, this forming a

minimal economic submodel that is able to display an energy transition from fossil- and biomass-based to renewables-based production. Processes we drop are population dynamics, migration, and knowledge spillovers.

- On the finer level, each social system possesses just two "cells", a "boreal" and a "temperate" cell in the "global North", and a "tropical" and a "subtropical" cell in the "global South", all of which differ in their initial fossil stocks and solar insolation. The environmental processes (ENV) we keep are a simplistic carbon cycle in the "planet" box interacting with a simplistic vegetation model in the four cells.

- Finally, to be able to represent social dynamics interacting with the Earth system, the model has a representative sample of 100 individuals connected by a social network. The only socio-cultural processes (CUL) we keep are the social learning of environmental friendliness driven by differences in well-being, and the voting on energy policy, this forming a minimal feedback loop between economy and policy.

- The resulting model hence contains a minimal set of processes forming a feedback loop that spans the socio-cultural, socio-metabolic, and environmental spheres, and allows us to get an idea of how much difference the inclusion of the socio-cultural sphere in Earth system models can make. For this, we performed a bifurcation analysis that varies the overall strength of the socio-cultural processes. This analysis shows a transition between different regimes and provides evidence that the strength of socio-cultural processes has a nontrivial influence on the trajectory of the Earth system beyond what can be represented by simple exogenous emissions scenarios.

**Editor's summary**

*This is a timely, ambitious, and important contribution to the scientific debate about the role of human actions on the Earth system. While the authors have made improvements to the manuscript, I have concluded that it is not currently acceptable for publication based on the latest round of reviews and a consideration of previous reviews and how the manuscript has developed. Consequently I recommend revisions with another round of peer review. I am confident that this review process could be completed quickly and that this manuscript can be published in ESD in a timely manner.*

*I see three main possible contributions of the manuscript:*

*C1 - Argue that coupling of human actions on the Earth systems is necessary or important for a class of scientific and policy-relevant questions.*

*C2 - Assume C1 (coupling is required) and then demonstrate how such coupling can be implemented into a scalable framework.*

*C3 - Provide worked examples of the output of a coupling framework that provides new scientific insights and findings that may be of policy relevance.*

*In attempting all three contributions, the manuscript is at risk of not comprehensively addressing any. In its current guise the manuscript attempts to justify the requirement for coupling, show how it could be achieved with a particularly modelling or implementation framework, and then demonstrate output from this implementation. This is ambitious. My questions on publication surrounds whether its feasible, or even desirable, to do all of that in a single publication. The temptation is to ask for more in order to cover C1-C3, but this may not be the best approach.*

*Given the authors do not present a single model, but a framework with which a potentially very large range of models could be developed and then integrated, the issues of reproducibility are quite complicated.*

*Documentation of the software implementation of the python copan:CORE model is available at the GitHub site. There is also documentation available at [https://pycopancore.readthedocs.io/en/latest/index.html](https://pycopancore.readthedocs.io/en/latest/index.html)*

*This includes a set of tutorials, e.g. how does a user uses the package. It also describes some of the implementation of the assumptions behind the model. I take the job this (or similar) documentation also needs to do, is establish C2. Presentation of a model framework (i.e. code that 'does something' with instructions of 'how to do something' does not necessarily establish that C2 has been satisfied.*

*In these respects, what is absent is software verification and validation. This is what I take to be the substantive issue of journal scope. ESD does not require nor does it request the presentation of the verification and validation of software that is used to produce scientific output. ESD would rely on the this activity being done in another journal, platform, or forum. From an editorial perspective, this has been one of the main challenges of ensuring appropriate review of the manuscript.*

*To proceed, I suggest the following routes to revision. Consider a clear separation between the model implementation and the assumptions surrounding it and its output. Specifically:*

*R1 - Argue that there is a need - scientific and of policy relevance - for such coupling and then*

*R2 - that the overall design approach taken - in particular the use of agent based implementations - satisfies this need.*

*R3 - Demonstrate the utility of this approach with results that either could not be produced via 'traditional' or established models and frameworks, or results that provide new insights into the class or perhaps example of the scientific (and potentially policy relevant) questions that the authors are motivated to address.*

*The existing manuscript does currently address R1-R3, but there are some gaps. For example, it may be argued that (potentially computationally expensive) agent based implementations are not required for some of the class of scientific questions that the authors argue their framework is designed for. There is a broader issue of to what extent human decision-making and behaviour needs to be disaggregated in social-ecological models. It is not the author's job to conclude such arguments! But the authors should be mindful that some readers (and reviewers) may need to carried along with their argument that is there is utility in agent based implementations.*

*Currently, for R1 a good deal of the 'intellectual heavy lifting - the 'motivation' comes from just two papers Verburg et al., 2016; Donges et al., 2017a, b and five bullet points in section 1.2.2 Design principles for World-Earth models. Given the planetary-scale ambition of the authors, it is understandable that there cannot be a sufficient list of possible scenarios and scientific questions that the framework will seek to address. But the absence of a discrete set of questions, plus some opacity as to the central modelling assumptions means that R1 is not sufficiently addressed.*

*For R2, it should be sufficient to focus on the disaggregated social dynamics approach - that is to establish that at a planetary-scale, aggregating human behaviours is not sufficient as it means missing important dynamics. I would have assumed that a planetary-scale framework would have some sort of tunable parameter by which this level of disaggregation can be 'turned up/down' as appropriate. Arguing that it's important to include low(er) level dynamics in order to understand high(er) level behaviours/states/responses works inversely: that is, it is sometimes better (in terms not*

*just of computational cost/efficiency but also in terms of mathematical tractability and conceptual understanding) to simplify lower level dynamics. Sometimes more is less.*

*For R3 you have already presented output from core:COPAN. This is 'toy' model output as I take your main motivation is to demonstrate that the core:COPAN framework works - it is able to produce model output. I do not think it is necessary to implement a real world model (to use empirical data for the choice of model elements and parameter values), but I do think that there could be more convincing results presented. It is not surprising that a more complex model will produce quite different results. What new insights, what new knowledge can be gained here? This questions can be addressed using a toy or conceptual model. I would hope, that once the manuscript is freed from the requirement of establishing that the copan:CORE framework is verified and validated, you would be able to concentrate on demonstrating some of its power. This would not require large numbers of new simulations and analysis.*

*In this way, the manuscript is bookended by R1 and R3. R3 in an important sense helps demonstrate R1 - why do we need these potentially very complex models and approaches?*

*Separate (and perhaps in parallel) to this, I would recommend you expose the modelling framework to suitable review with verification and validation built into that process via perhaps a dedicated modelling journal. This more modular approach would allow you to produce a series of publications using a common and well established foundation for the implemented framework which could be referred and referenced to accordingly.*

*If you are able to revise your manuscript, then I would seek an additional round of independent peer review - but of a potentially limited nature. While I do not want to prejudice future outcomes, it may be sufficient for the review to progress with the response of one of the reviewers from the last round of peer review. I do not think it would be necessary to effectively treat the manuscript as a new submission with an entirely new round of peer review. However, I am open to your alternative suggestions.*

*Thank you for your continued contribution to Earth System Dynamics.*

**Referee 1 comments on revised manuscript copan:CORE**
**(Brian Dermody)**

*I am pleased to comment favourably on this revised manuscript. The authors have made considerable efforts to improve the manuscript. The revised manuscript now*

*clearly outlines the motivation for a new generation of WEMs, the theoretical foundation for their framework and how they have implemented it. I hope that the authors will be able to successfully develop what they present here. That undertaking will make an urgent and important contribution to understanding the complex drivers of social and environmental change in our globalised world.*

*Specific comments*

*The revised structure for sections 1.1 and 1.2 is much improved and clearly outlines the current state of modelling earth system processes, the shortcomings of these approaches and the motivation for the new framework presented here as well as a helpful introductory overview to the theories and concepts relevant for WEMs. The addition of a rationale section is a great help for helping the reader understand the motivation and relevance of model design choices. I think it is a good decision to move most of the detailed code description to the SI and point readers to the most recent API documentation online. I agree with the editor on this that the discussion manuscript was somewhat caught between framework specification which might be more appropriate for a software journal such as Geoscientific Model Development and an introduction and academic motivation of your modelling framework, which I think is what you want to achieve here. The link between abstract concepts of culture, social-metabolism and environment in figure 2 with figures 3 and 4 is a big help in linking there theoretical grounding for your framework with how it will actually be implemented.*

*Section 1.2.2 on design principles is much improved and provides a powerful motivation for a new generation of WEMs.*

> Motivated by this assessment, we tried to improve it further to make an even stronger case.

*The new section 2.3.5 is a welcome addition, which clarifies how the model may serve to couple internal and external model components.*

> We think so, too, and kept this, but we had to move it to the SI
> to accommodate the new overall design of the manuscript agreed with the editor.

*Page 9, line 16: Perhaps provide the link to the git here.*

> We have included that link in the manuscript now.

***Referee 3 comments on revised manuscript***

*The manuscript by Donges et al deals with a highly timely and, I think, innovative approach to include human societies into an Earth system context, and thus the topic is certainly suitable for publication. With this manuscript, however, I have some problems that prevent me from recommending acceptance.*

*What I find difficult is that the paper lacks a clear focus. It describes the challenge associated with bringing socio-cultural dynamics into an Earth system model, a modelling approach, and an application. I think that each of these points are potentially valid and scientifically challenging topics, but the level at which these are dealt with in the manuscript is rather shallow and I feel that not much can be learned from it. So I think it needs a major revision.*

*To start with, why is it important to couple socio-cultural dynamics with the biophysical Earth system dynamics? There should clearly be a dividing line for topics that require such kind of model, and other topics that do not. It reminds me of an equivalent question that regards the coupling of atmosphere and ocean models. Not all studies require an interactive ocean model to run, and simply using prescribed boundary conditions in terms of sea surface temperatures is for quite a range of topics fully sufficient, and, in fact, quite often also more realistic because it does not allow the atmospheric model to drift as much. So I imagine that something similar would apply to the coupling of socio-cultural dynamics and the Earth. But this manuscript does not contain much about the conceptual challenge and the topics for which such a model would be needed and for which it would not be needed. Such a clearer conceptual description, best illustrated with a concrete example, I think would really improve the manuscript.*

> We agree with the referee and expanded the discussion of more concrete research questions for WEM and added a discussion of questions that do not require such a coupling in Section 1.

*Second, a thorough model description should enable a reader to reproduce the model. At the moment, this manuscript would not allow this level of reproducibility. It perhaps does not need to be provided at that level, because if it would describe it at this level, the manuscript would be more suitable for a model description journal like GMD. At the moment, the level of model description provides some parts that I think are probably too detailed, but on the other hand, it does not allow me to understand (or reproduce) the example provided in section 3 of the manuscript. For instance, what are the major assumptions, what are the critical parameters and the key uncertainties?*

As agreed with the editor, we have now simplified the example model, moved the model detail from the SI to the main text, and added missing details. In addition, all model code is freely available on the git repository.

*I may also point out that I do not think that neither an object-oriented framework nor an agent-based approach are essential to reproduce human dynamics. It is fine to use this, but I am sure one could equally represent these dynamics with conventional differential equations. Even more so, the use of agent-based dynamics and stochasticity may introduce an element of randomness which could impact the extent to which the results are reproducible, which would be a big problem. There are sufficient number of examples in the ecological literature where spatial population dynamics are represented with differential equations. So I think it is important to be clear about separating the conceptual challenge from the implementation.*

We believe there are several aspects to this.

We agree that the question of whether or not to use an object-oriented framework is not determined by the system one wants to model. Our reason for choosing it is rather a methodological one: since we argue World-Earth modeling is an inherently interdisciplinary challenge that needs to use a language and framework accessible to several communities, we believe an object-oriented approach is the most natural way of thinking about the system, as nearly all disciplines' verbal language suggests the existence of entities of different types, possessing attributes that may change due to processes.

Regarding the use of agent-based model (ABM) components, we remain agnostic but argue that this is a technique a sufficiently open framework should allow among other approaches. Whether or not all ABMs can just as well be described (or at least their average behaviour approximated, e.g. methods from statistical physics) by ordinary differential equations remains an open question that we do not have to solve here. Other works by us make heavy use of macroscopic approximations, so we do not want to promote ABMs here but rather enable their use where the modeler deems them appropriate, e.g. when representing a large degree of heterogeneity and social structure is important to certain research questions . We have therefore reduced their mentioning in the text now and also argue that the modular design of copan:CORE explicitly supports comparison studies between ABM-based and ODE-based versions of a model.

[revised manuscript text omitted]

*Third, the example provided in section 3 sounds interesting, but there is far too little information provided for me to understand what is happening and why.*

> As mentioned above, we now explain and discuss that model in much more detail now.

*So overall, I think the manuscript would be much improved (and suitable for publication) if it strengthened the conceptual challenge, provides a description of the key processes that are needed to describe the example in section 3 in the methods, and then describe the results of the example in much greater detail. The present version of the manuscript I find lacks the depth and details to make it a scientifically reproducible contribution, and it misses to more clearly describe the conceptual challenge associated with bringing human dynamics into Earth system modelling.*

> We address these points in our second revision of the manuscript as we have summarized in our response to the editor's comments above.

---

## Author Response (AR3)

**Response letter, 3[rd] revision of manuscript "Earth system modeling with endogenous and dynamic human societies: the copan:CORE open World-Earth modeling framework"**

Following the editor's recommendations (communicated on Dec 24 2019 via email) below (quoted in *italics* below), we have produced a third revision of the manuscript. Changes in the text and updates and additions in the references in the paper and SI text are highlighted in red colour therein.

> *I've read through the latest version today and the reviews and responses. [...]*
>
> *There is the scope to seek to revise the paper in some considerable respects. Perhaps even reverting some of it back to a previous version. But I don't think that would be in anyone's interests. I have the following suggestions/observations based on the latest version.*

We thank the editor for this assessment and give a point-to-point reponse to his recommendations in the following:

> *1 - An important motivation is to promote copan:CORE to potential users. Rather than a "towards" paper, a paper that is in some sense a manifesto that argues that "something should be done", your manuscript alerts researchers to a new thing in the modelling world and asks them to use it. I would assume it would be critical that users are able to quickly and effectively get to grips with copan:CORE. I still get a 404 for the link http://pycopancore.readthedocs.io/ on the GitHub page. The only documentation I can see is very limited. This would also in part address some of the grumbles about reproducibility.*

1 - Thank you for pointing this out. We have fixed this error now and the full API documentation of copan:CORE is now online and accessible via github.

> *2 - There will always be some pointed skepticism about the role, place and utility of global socio-ecological models (e.g. World3). I wouldn't shrink from addressing some of them quite directly. Be open about that fact that some people/communities think they are very poor science. I think the power of your approach is that you are agnostics about modules. The proof will be in the pudding in that individual model formations will sink or swim on their robustness. You are providing the platform for model development. For the motivation of copan:CORE I think it's sufficient to point out that many of the key sustainability challenges we face intimately involve the sort of coupled dynamics and feedback loops that copan:CORE is designed to implement effectively.*

2 - We agree that it is important to mention these concerns and general challenges for integrated human-Earth system modeling more explicitly. We have added a corresponding paragraph to the introduction (at the end of Subsection 1.2). We have furthermore added references to several recently published papers that further strengthen the case for copan:CORE and World-Earth modeling and concretely discuss corresponding challenges and approaches to address them in much more detail (e.g. Calvin and Bond-Lamberty 2018; Beckage et al. 2018; Barton and the Open Modelling Foundation 2019; Schill et al. 2019 etc.). These and other papers really show that a growing community is moving in this direction and we hope that our article could bring this to the attention of ESD readers as well.

> *3 - But beyond offering a platform, I think you are also potentially looking at managing a community of modellers or at least a repository of copan:CORE models. How that is managed will I think be crucial. It would be great to see enthusiastic take up of the platform and many models being produced. But it would a missed opportunity if these models were not curated in some respect. We need to ensure the sum of the modellers efforts are greater than the parts. We need ways to share best practice for these sorts of models. Perhaps some copan:CORE tailored practices from Software Carpentry (https://software-carpentry.org) perhaps? I appreciate that is potentially way beyond the scope of the paper, but if this paper is the first exposure someone has to the platform then you will want some things in place to ensure you "capture" them and their outputs.*

3 - We now discuss in more detail this community building challenge and how copan:CORE is specifically designed to facilitate that in the conclusions (Section 4).